# Dual roles for ATP in the regulation of phase separated protein aggregates in *Xenopus* oocyte nucleoli

Michael H Hayes[1], Elizabeth H Peuchen[2], Norman J Dovichi[2], Daniel L Weeks[1,3]*

[1]Molecular Medicine Doctoral Program, University of Iowa Carver College of Medicine, Iowa City, United States; [2]Department of Chemistry and Biochemistry, University of Notre Dame, Notre Dame, United States; [3]Department of Biochemistry, University of Iowa Carver College of Medicine, Iowa City, United States

**Abstract** For many proteins, aggregation is one part of a structural equilibrium that can occur. Balancing productive aggregation versus pathogenic aggregation that leads to toxicity is critical and known to involve adenosine triphosphate (ATP) dependent action of chaperones and disaggregases. Recently a second activity of ATP was identified, that of a hydrotrope which, independent of hydrolysis, was sufficient to solubilize aggregated proteins in vitro. This novel function of ATP was postulated to help regulate proteostasis in vivo. We tested this hypothesis on aggregates found in *Xenopus* oocyte nucleoli. Our results indicate that ATP has dual roles in the maintenance of protein solubility. We provide evidence of endogenous hydrotropic action of ATP but show that hydrotropic solubilization of nucleolar aggregates is preceded by a destabilizing event. Destabilization is accomplished through an energy dependent process, reliant upon ATP and one or more soluble nuclear factors, or by disruption of a co-aggregate like RNA.
DOI: https://doi.org/10.7554/eLife.35224.001

**\*For correspondence:**
daniel-weeks@uiowa.edu

**Competing interests:** The authors declare that no competing interests exist.

## Introduction

Maintenance of protein solubility is crucial for proteostasis. Dysfunction can lead to the aggregation of functional soluble proteins into pathologic self-templating complexes called amyloid. Amyloid aggregates underlie the cytotoxicity seen in Alzheimer's disease, type II diabetes, amyotrophic lateral sclerosis, and many other pathologic conditions (*Knowles et al., 2014*). However, the coalescence of proteins into macromolecular complexes, which undergo liquid-liquid phase separation (LLPS) (*Banani et al., 2016*; *Berry et al., 2018*; *Courchaine et al., 2016*; *Lee et al., 2013*) or self-assemble into amyloid fibrils, is vital to a diverse set of cellular processes. These aggregated proteins contribute to activities such as hormone storage, formation of non-membrane bound organelles, and long-term memory (*Courchaine et al., 2016*; *Boke et al., 2016*; *Heinrich and Lindquist, 2011*; *Maji et al., 2009*; *Majumdar et al., 2012*). Many proteins naturally aggregate. For example, some proteins with RNA binding or low complexity domains exist in both soluble and aggregated states. Switching from one state to the another can be controlled by concentration, environment, or presence of a co-aggregate. In other instances, the switch is an energy dependent process (*Phair and Misteli, 2000*; *Shorter and Lindquist, 2004*).

Depletion of adenosine triphosphate (ATP) has been shown to influence the balance between soluble and aggregated states for proteins in yeast and metazoans, suggesting normal proteostasis is maintained by energy dependent chaperones and disaggregases (*Lee et al., 2013*; *Brangwynne et al., 2011*; *Feric et al., 2016*; *Hyman et al., 2014*; *Kroschwald et al., 2015*; *Saad et al., 2017*). These enzymes, such as yeast Hsp104 or the Hsp70 co-chaperone complex in

metazoans, rely on ATP hydrolysis to maintain the dynamic nature of functionally aggregated proteins and reverse pathological aggregation (*Kroschwald et al., 2015*; *Saad et al., 2017*; *Nillegoda et al., 2015*; *Shorter, 2011*; *Vacher et al., 2005*).

ATP, best known as the energy currency of the cell, has recently been shown to act as a hydrotrope to destabilize aggregated proteins (*Patel et al., 2017*). Hydrotropes are co-solvents capable of bringing poorly soluble or insoluble compounds into solution (*Eastoe et al., 2011*). Hydrotropic solubilization is concentration dependent requiring that the minimum hydrotropic concentration of the hydrotrope be reached before they act as a co-solvent (*Eastoe et al., 2011*; *Schreier et al., 2000*). Above the minimum hydrotropic concentration, hydrotropes promote solubilization by decreasing the interfacial tension between solute and solvent. Some hydrotropes seem to assume micellular structures while others facilitate solubilization through the formation of more open, planar structures (*Schreier et al., 2000*). Meyer and Voigt, (*Meyer and Voigt, 2017*) in discussing the possible role of ATP as a hydrotrope, point out that ATP would not be expected to form micelles but does have the chemical features needed to serve as a hydrotrope. ATP is able to function as a hydrotrope because it is an amphiphile, a compound with hydrophilic and hydrophobic properties. For ATP, the adenosine ring contributes a hydrophobic moiety while the 5' triphosphate contributes the hydrophilic component.

Using recombinant protein in vitro, Patel and colleagues demonstrated that the hydrotropic activity of ATP was sufficient to solubilize LLPS droplets made from purified protein at concentrations approaching ATP's maximum physiologic level (4–8 mM). In contrast, lower yet still physiologic, concentrations of ATP (1–2 mM) had a limited effect on protein solubility. Patel et al. (*Patel et al., 2017*) postulated that high intracellular concentration of ATP contributes to maintenance of protein solubility and inhibits pathologic protein aggregation in an energy independent manner (*Patel et al., 2017*). How interactions with other proteins, post translational modification, or the presence of co-aggregates, like RNA or poly-ADP ribose, may influence an endogenous aggregate's susceptibility to hydrotropic action was not investigated (*Saad et al., 2017*; *Andersen et al., 2005*; *Boamah et al., 2012*; *Ellis et al., 2010*; *Kar et al., 2011*; *Tollervey and Kiss, 1997*). Among the cellular structures that feature aggregated proteins are non-membrane bound organelles. For example, nucleoli are formed by the coalescence of proteins into functional aggregates, some of which demonstrate liquid-like behavior, yet contain an amyloid-like component (*Brangwynne et al., 2011*; *Feric et al., 2016*; *Hayes and Weeks, 2016*). These features make nucleoli an ideal tissue to test the ability of ATP to act as hydrotrope on aggregates of proteins in vivo.

Nucleoli are centers for rRNA synthesis, processing, modification, and ribosome-subunit assembly. They have 100s of proteins, and multiple classes of RNA, including rRNA precursors, mature rRNA species and many small nucleolar RNAs (snoRNAs). RNA metabolism plays a central role in nucleolar formation, maintenance, and retention of nucleolar components (*Hayes and Weeks, 2016*; *Franke et al., 1981*; *Verheggen et al., 2000*). De novo nucleolar formation is dependent upon the coalescence of RNA binding proteins, such as fibrillarin (Fbl), nucleolin, and nucleophosmin (Npm1), around pre-made 40S rRNAs (*Verheggen et al., 2000*). For at least two of these proteins, Fbl and Npm1, rRNA helps to mediate protein aggregation and LLPS in vitro (*Feric et al., 2016*). The nucleolus is composed of three subregions, whose separation and immiscibility is believed to be the result of differing surface tension values between subregions (*Feric et al., 2016*). The subregions are the granular component, which is the site of ribosome assembly, the dense fibrillar component where rRNA processing and modification occurs, and the fibrillar core, which is the site of rRNA transcription. Each subregion is readily identified by specific markers. Npm1 is found within the granular component, while Fbl marks the dense fibrillar component, and the fibrillar core can be identified by the presence of rDNA repeats or RNA polymerase I (*Feric et al., 2016*).

The contents of the vertebrate nucleus, including proteome and pool size of various compounds including ATP, has been reported (*Wühr et al., 2015*; *Andersen et al., 2005*; *Gall et al., 2004*; *Kiseleva et al., 2004*). Most cells have one or a few nucleoli. *Xenopus* oocyte nuclei contain hundreds of functional nucleoli, some of which are larger than entire eukaryotic cells (*Gall et al., 2004*). Their large size is amenable to visual assays, and there is specific information regarding proteins that participate or interact with aggregated nucleolar proteins (*Brangwynne et al., 2011*; *Feric et al., 2016*). Using *Xenopus* oocyte nuclei, we developed a novel system to investigate the capacity of ATP to act as a hydrotrope on endogenous aggregates. We assembled a roster of proteins that form or are tightly associated with aggregates in the nucleus and nucleolus. Next, we identified

conditions where the nuclear environment can be manipulated to vary ATP concentration and alter the relationship between soluble and aggregated proteins. This allowed us to ask specific questions using naturally occurring aggregates in their endogenous, or near endogenous, environment. Under some conditions ATP and a non-hydrolysable analog, 5′-(β,γ-imido)triphosphate (AMP-PNP) behave similarly, supporting an energy independent role for ATP in solubilization of aggregated nucleolar proteins. However, this process requires a sensitization step, such as aggregate destabilization by RNA depletion or an energy dependent process that requires one or more soluble factors. These results support a two-step model of in vivo solubilization of poteins that coalesce to make aggregates. We propose that ATP initially acts as an energy source for a destabilizing step conferring susceptibility to solubilization, and is supported by the energy independent action of ATP as a biological hydrotrope.

## Results

### Aggregated proteins are selectively retained in isolated nuclei

We previously demonstrated that freshly isolated *Xenopus* oocyte nuclei contain ordered protein aggregates within many sub-nuclear organelles (*Hayes and Weeks, 2016*). In contrast to soluble proteins, these aggregates were held in place by the gelled nuclear actin meshwork that forms following isolation in aqueous buffer (*Wühr et al., 2015*; *Kiseleva et al., 2004*; *Paine et al., 1992*). To compare the retention of soluble and insoluble nuclear components, we injected mRNA encoding dTomato with a nuclear localization signal into stage V-VI oocytes (*Figure 1A*).

Following overnight incubation nuclei were manually isolated in OR2 buffer (*Figure 1B*), immediately stained with the amyloid dye thioflavin T, and imaged at 15 min intervals for 1 hr (*Figure 1C*). Robust dTomato fluorescence and thioflavin T staining was observed immediately post isolation. However, consistent with reports by others, dTomato fluorescence rapidly decreased and was at our limit of visual detection by 30 minutes (*Wühr et al., 2015*; *Paine et al., 1992*). In contrast, thioflavin T staining remains throughout this time course, indicating that aggregated proteins are retained even as soluble nuclear components diffuse away. The fate of small molecules, like ATP (MW = 507.18), was tested by injecting a fluorescent surrogate, Lucifer Yellow (MW = 521.57), into nuclei of stage VI oocytes. Nuclei were rapidly isolated in OR2 and imaged at 1 min intervals. The majority of Lucifer Yellow was lost by 5 min post isolation (*Figure 1G*), indicating small molecules rapidly equilibrate between isolated nuclei and their surrounding buffer. From an experimental design point of view, this allows for manipulation of the concentration of small molecules, like ATP, through supplementation of the isolation buffer. The concentration of small molecules in the nucleus will match the concentration in aqueous isolation buffer.

The endogenous protein content of isolated nuclei was analyzed via SDS-PAGE. Nuclei were collected and analyzed at 15 min intervals post isolation. Protein loss was easily apparent at 15 min, and near completion 45 min post isolation (*Figure 1D–E*). The retained protein pool is distinct from the initial protein pool (arrows, *Figure 1E*), and revealed enrichment of actin when compared to total protein (arrowhead, *Figure 1E–F*). Consistent with previous reports, densitometric analysis revealed that approximately 10% of the initial nuclear protein pool was retained after 1 hr of soluble protein depletion (*Figure 1E–F*) (*Paine et al., 1992*). This analysis demonstrates that most endogenous proteins are lost on the same timescale as soluble dTomato, indicating they too are soluble, and that incubation of nuclei for 1 hr provides a simple method to enrich for endogenously aggregated nuclear proteins.

To identify components of the retained protein pool, we isolated stage V-VI *Xenopus* oocyte nuclei in bulk, depleted them of soluble proteins and analyzed them by tandem mass spectrometry (MS/MS) (*Peuchen et al., 2016*; *Scalenghe et al., 1978*). This analysis identified over 120 proteins that are retained in soluble protein depleted nuclei (FDR = 0.01). Gene ontology analysis (GO::Term-Finder) revealed that many of the retained proteins are involved in ribonucleoprotein complex biogenesis, DNA replication, RNA processing and DNA-templated transcription (representative proteins are shown in *Table 1*, the full listing is in *Table 1—source data 1* and a key to all data sets is in *Table 1—source data 2*) (*Boyle et al., 2004*). Among the nucleolar proteins enriched in the retained aggregate enriched proteome were Nucleophosmin and Fibrillarin.

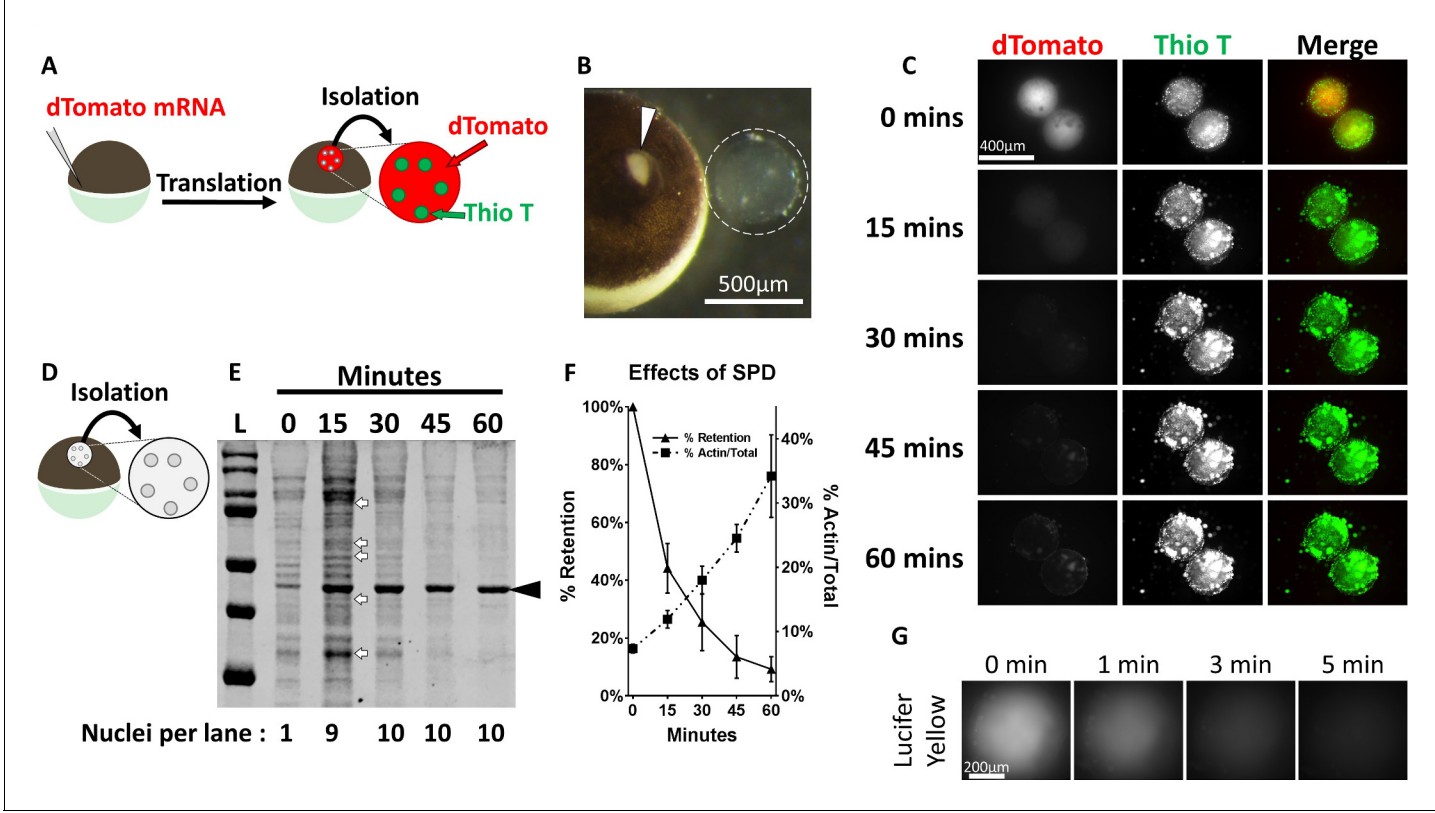

**Figure 1.** Protein aggregates are selectively retained in isolated oocyte nuclei. (**A**) Synthetic mRNA encoding nuclear localized dTomato was injected into stage V-VI *Xenopus laevis* oocytes 1 day prior to isolation. Following isolation nuclei were incubated in OR2 buffer for the indicated times then assayed. (**B**) Stage VI oocyte with incision site (arrowhead) and manually isolated nucleus (dashed circle). (**C**) 1 hr time-lapse images of aqueously isolated and Thioflavin T (Thio T, green) stained nuclei from dTomato-NLS (red) expressing stage VI oocytes demonstrates loss of soluble dTomato, but retention of Thioflavin T positive aggregates. (**D–F**) Nuclei were isolated from un-manipulated oocytes, incubated in OR2, collected at 15 min intervals, and analyzed by SDS-PAGE (**D**). Coomassie staining (**E**) with quantitation (**F**) of soluble protein depleted nuclei demonstrates rapid loss of soluble endogenous proteins and retention of aggregate associated proteins. The number of nuclear equivalents per lane is indicated at the bottom of (**E**). Arrows highlight the subset of proteins enriched following depletion of soluble proteins. Arrowhead highlights 42 kDa actin, which is enriched following soluble protein depletion (**F**). (**G**) Time-lapse images of an isolated stage VI oocyte nucleus immediately following nuclear injection of Lucifer Yellow, a fluorescent ATP surrogate. Images in (**C**) and (**G**) are representative from two independent experiments encompassing at least six nuclei. Data in (**F**) contains three biological replicates (3–10 nuclei per replicate) representing material from two separate frogs.
DOI: https://doi.org/10.7554/eLife.35224.002

Nucleophosmin (Npm1), localizes to the nucleolar granular component and Fibrillarin (Fbl) is found within the nucleolar dense fibrillar component (*Figure 2A*). Both proteins can form aggregates, and have been highlighted as proteins that coalesce into phase separated liquid droplets (*Feric et al., 2016*). We verified their retention by injecting GFP-Npm1 and RFP-Fbl encoding mRNA into stage V-VI oocytes (*Figure 2B*). After allowing fluorescent proteins to accumulate overnight, we manually isolated nuclei and examined GFP-Npm1 and RFP-Fbl localization using fluorescent microscopy (*Figure 2C*). Retention of these proteins was confirmed and quantified via western blot analysis of oocytes injected with GFP labeled Npm1 or Fbl, along with nuclear localized GFP as an internal soluble protein control. 20.8% ± 8.7 of GFP-Fbl and 24.9% ± 11.2 of GFP-Npm1 (*Figure 2D* and *Figure 2—figure supplement 1*) were retained after 1 hr of soluble protein depletion.

Compared to total nuclear protein, Npm1 and Fbl were enriched (8.60 ± 5.72 fold enrichment of Fbl and 7.79 ± 2.45 of Npm1, *Figure 2E* and and *Figure 2—figure supplement 1*) in the retained pool. Like dTomato (*Figure 1*), nuclear localized GFP behaves like other soluble proteins and was present when nuclei were isolated and assayed within 15 minues. However, following soluble protein depletion roughly 1.0% of control GFP was retained, representing a 0.09 ± 0.03 fold enrichment compared to a total nuclear protein. This is consistent with previous reports investigating the

**Table 1.** Representative listing of aggregate and aggregate associated proteins in oocyt nuclei.

| Process | Gene name | Function (summarized from www.genecards.org) |
|---|---|---|
| DNA Replication | | |
| | TOP2A | Topoisomerase II Alpha, DNA topoisomerase |
| | MCM2 | Minichromosome Maintenance Complex Component 2, involved in replication initiation |
| | POLE | DNA Polymerase Epsilon, Contributes to DNA repair and replication |
| | WRN | Werner Syndrome RecQ Like Helicase, DEAH 5' to 3' DNA with 3' to 5' exonuclease activity |
| | PRIM2 | Primase Subunit 2, DNA directed RNA polymerase that generates lagging |
| RNA Processing | | |
| | GTF2H2 | General Transcription Factor IIH Subunit 2, member of RNA pol II transcription initiation factor IIH complex |
| | SRRM1 | Serine and Arginine Repetitive Matrix 1, Part of pre- and post-splicing multiprotein mRNP complexes |
| | XAB2 | XPA Binding Protein 2, participates in mRNA splicing and nucleotide excision repair |
| | NOLC1 | Nucleolar and Coiled-Body Phosphoprotein, heavily modified regulator of RNA pol I |
| | SRSF5 | Serine Arginine Rich Splicing Factor 5, spliceosome component with an RRM and RS domains |
| Transcription, DNA-templated | | |
| | POLR1C | RNA Pol I subunit C, component of RNA Pol I and III complexes |
| | HSPA8 | Heat Shock Protein Family A Member 8, constitutively expressed member of HSP70 family |
| | NCL | Nucleolin, binds histone H1 to decondense DNA |
| | POLR1A | RNA pol I subunit A, catalytic and largest subunit or RNA Pol I complex |
| | POLR2E | RNA Pol II subunit E, subunit shared between RNA Pol I, II, and III |
| Ribonucleoprotein Complex Biogenesis | | |
| | PES1 | pescadillo, member of PeBoW complex and contributes to ribosome biogenesis, binds BRCA1 |
| | BOP1 | Block of Proliferation 1, participates in rRNA processing and is a member of PeBoW complex |
| | NPM1 | Nucleophosmin, multi-functional nucleolar phosphoprotein |
| | DDX21 | DEAD-Box Helicase 21, facilitates processing or RNA Pol I and II transcripts |
| | FBL | Fibrillarin, rRNA and protein methyltransferase |

DOI: https://doi.org/10.7554/eLife.35224.003

The following source data available for Table 1:

Source data 1. Retained proteins following depletion of soluble nuclear components.
DOI: https://doi.org/10.7554/eLife.35224.004

Source data 2. Key with brief descriptions of the data sets.
DOI: https://doi.org/10.7554/eLife.35224.005

dynamics of Npm1 and Fbl which showed they exist in both soluble and aggregated states (*Boke et al., 2016*).

With these observations, we established the conditions used to examine isolated *Xenopus* oocyte nuclei as a simple model system for studying endogenous nuclear aggregates in vitro (after 1 hr of soluble protein depletion), in situ (freshly isolated and immediately assayed while soluble proteins are still present though small molecule concentrations equilibrated to match buffer conditions), or ex vivo (isolated under oil, allowing both nuclear proteins and small molecules to be retained [*Handwerger et al., 2005*; *Lund and Paine, 1990*]).

## Aggregated nucleolar proteins are resistant to ATP mediated hydrotropic solubilization

Patel and colleagues demonstrated that ATP and its non-hydrolysable analogue adenosine 5'-(β,γ-imido)triphosphate (AMP-PNP) could act as hydrotropes to solubilize recombinant liquid-liquid phase separated protein droplets (*Patel et al., 2017*). Our initial assays were carried out using 8 mM ATP because it is similar to the amphibian nuclear concentration of 6.2 ± 0.7 mM (*Miller and*

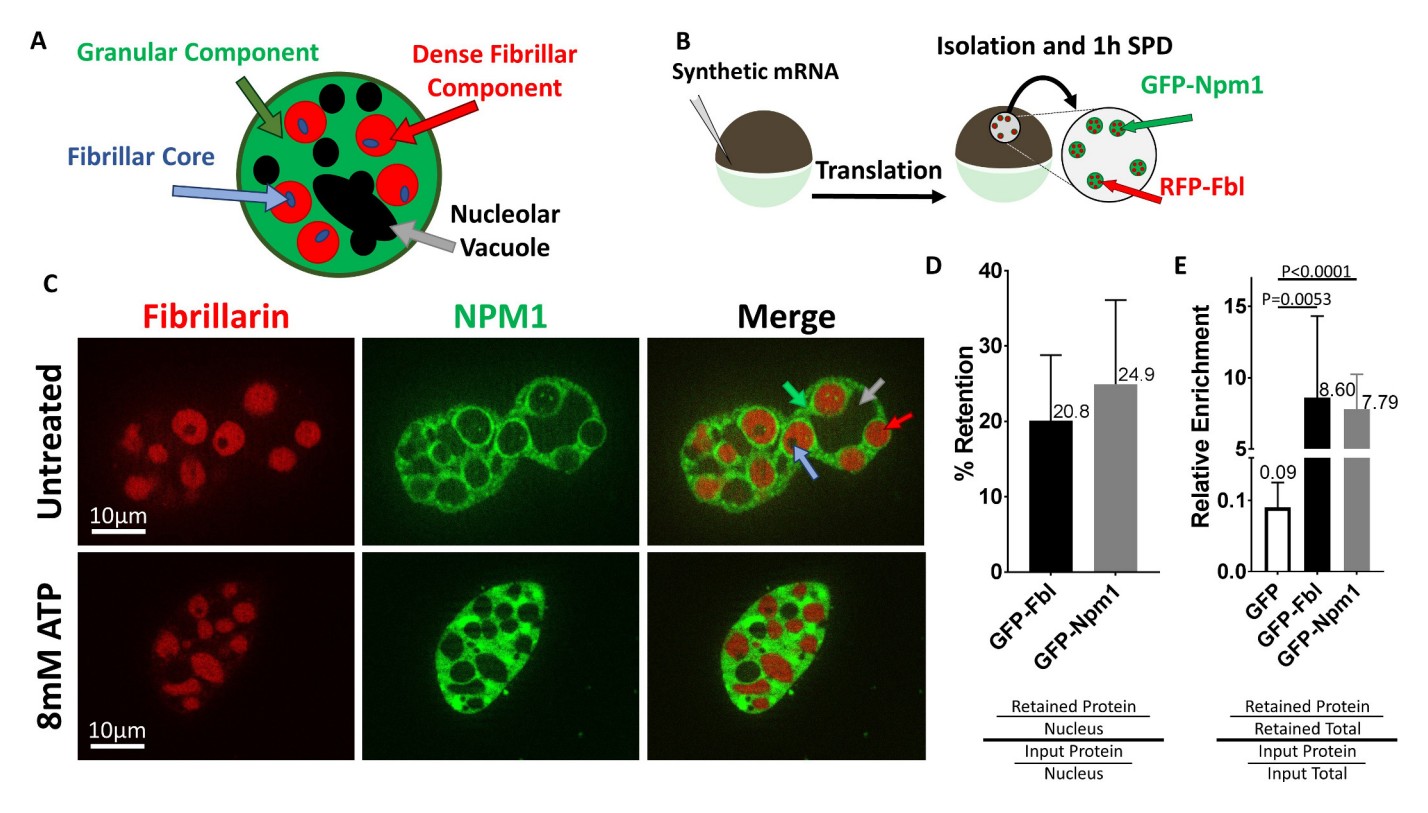

**Figure 2.** Nucleoli are resistant to ATP mediated hydrotropic solubilization. (**A**) Schematic diagram of tri-laminar nucleolar architecture, with granular component in green, dense fibrillar component in red, and fibrillar core in blue. Nucleolar vacuoles are unlabeled and will appear black. (**B**) Synthetic mRNA encoding GFP-NLS or fluorescently labeled Npm1 and/or Fbl were injected into stage V/VI oocytes. Following overnight incubation nuclei were isolated, depleted of soluble proteins for 1 hr in OR2, then analyzed by fluorescent microscopy (**C**) or anti-GFP immunoblotting (**D–E**). (**C**) Npm1 (green) or Fbl (red) are retained (top) and resistant to solubilization by 8 mM ATP (bottom). (**D–E** and *Figure 2—figure supplement 1*) Nuclei from GFP-Fbl/GFP-NLS or GFP-Npm1/GFP-NLS co-expressing oocytes were isolated and depleted of soluble proteins (retained) and compared to GFP-Fbl or GFP-Npm1 levels in immediately harvested (input) nuclei (**D**) or total protein (**E**). Labeled proteins were analyzed via anti-GFP Western Blot, and total protein was determined via REVERT protein staining (LI-Cor). Schematic of methodology and representative raw data for (**D–E**) can be found in *Figure 2—figure supplement 1*. Images in (**C**) are representative optical sections captured with an Apotome equipped Zeiss Axioplan2 microscope, and do not have identical exposure times. Data in (**D–E**) is derived from 4 (GFP-Npm1 and GFP-FBL), or 6 (GFP) biological replicates (groups of 2–20 nuclei).
DOI: https://doi.org/10.7554/eLife.35224.006

The following figure supplement is available for figure 2:

**Figure supplement 1.** GFP-Fbl and GFP-Npm1, but not GFP-NLS, are Retained Following Soluble Protein Depletion.
DOI: https://doi.org/10.7554/eLife.35224.007

*Horowitz, 1986*), within the reported physiologic range of other eukaryote nuclei, and sufficient to solubilize recombinant LLPS droplets in vitro (*Patel et al., 2017*; *Miller and Horowitz, 1986*; *Traut, 1994*). Oocytes injected with mRNA encoding GFP-Npm1 and RFP-Fbl were used as a source of nuclei. Nuclei were isolated in OR2, depleted of small molecules and soluble proteins for 1 hr, then transferred into in situ buffer (ISB, which compared with OR2 provides better osmotic support, buffering to stabilize pH when adding nucleotides and potassium rather than sodium to better align with cellular ion concentrations) supplemented with 8 mM ATP or AMP-PNP for 15 min prior to imaging. Fluorescently labeled proteins were retained following this treatment (bottom row, *Figure 2C*).

Following the same treatment, no changes in thioflavin T staining, used to detect aggregates with amyloid content, were noted (Fig 5B). These findings demonstrate that, although effective on purified protein aggregates in vitro, restoring physiologic levels of ATP alone was not sufficient to solubilize naturally aggregated proteins. Nucleolar proteins are known to cycle between the nucleoplasm and nucleolus (*Chen and Huang, 2001*) so we considered whether soluble protein depletion

resulted in irreversible aggregate modification, conferring resistance to solubilization. Specifically, we sought to determine if both Npm1 and Fbl were still capable of exchanging between soluble and aggregated pools, represented by maintenance of the ability to transfer from one nucleolus to another. This assay takes advantage of the ability to fuse oocyte nuclei, isolated using conditions that deplete or retain soluble nuclear proteins, with one another. Once prepared, nuclei are fused under oil (*Figure 3A*).

We determined that proteins were exchanged between nucleoli originating from oil isolated nuclei singly expressing either Npm1 or Fbl reporters. These nuclei should have a full complement of soluble protein and native levels of ATP and other small molecules. When two oil isolated nuclei were fused together, proteins exchange between nucleoli as soon as the samples were placed on the microscope (*Figure 3C*), demonstrating how dynamic both Npm1 and Fbl leaving and rejoining their respective nucleolar aggregates can be. Subsequently, soluble protein depleted nuclei, from dually labeled oocytes, were transferred into light mineral oil and merged with unlabeled nuclei isolated under oil (*Figure 3A–B*). Within 15 min it was evident that aggregated Npm1 and Fib present

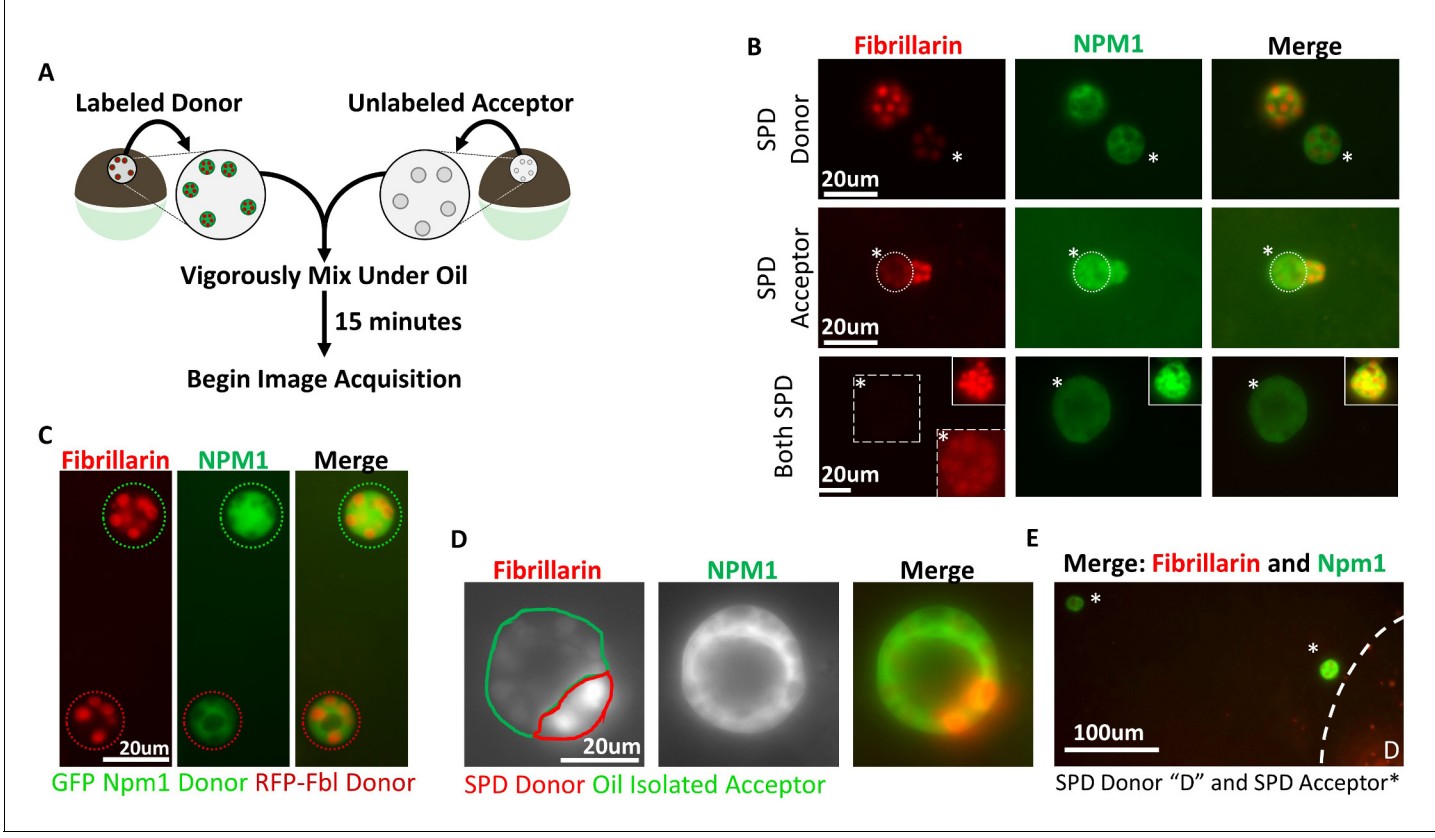

**Figure 3.** Aggregated nucleolar proteins retain intrinsic capacity for dynamic proteins exchange, but that capacity is enhanced by soluble nuclear components. (**A**) Donor nuclei were isolated from oocytes co-expressing GFP-Npm1 and RFP-Fbl, while acceptor nuclei were isolated from un-injected oocytes. (**B**) Top, soluble protein depleted (SPD) donor nucleus was mixed with oil isolated acceptor nucleus, demonstrating soluble protein depletion does not irreversibly alter nucleoli, and suggests that a soluble nuclear factor(s) is responsible for the dynamic nature of nucleoli. Middle, transfer of labeled protein from oil isolated donor to soluble protein depleted acceptor indicates that soluble protein depletion does not prevent recruitment nor aggregation of soluble proteins. Bottom, aggregated proteins exchange between soluble protein depleted nucleoli. Exposure was adjusted in bottom images to account for decreased signal, and donor nucleolus was included in top inset for comparison. Brightness and contrast were adjusted in dashed inset to aid in RFP-Fbl visualization. Asterisks indicates the acceptor nucleolus. (**C**) Exchange of fluorescently labeled proteins from oil isolated and merged nuclei singly expressing GFP-Npm1 or RFP-Fbl. (**D**) Trans-nucleolar fusion following mixing of soluble protein depleted RFP-Fbl expressing donor (red) and oil isolated GFP-Npm1 (green) under oil demonstrates rescue more normal rounded phenotype. (**E**) Exchange of GFP-Npm1 and RFP-Fbl is diffusion limited. As distance from donor 'D' increases fluorescence intensity of acceptor decreases. Images in (**B–C**) are representative of 3 independent experiments and at least nine biological replicates (merged nuclei). Images acquisition began 15 min after trans nuclear mixing. The fusion event depicted in (**D**) is the most complete event we observed. Partial fusions (e.g. middle panel B) are commonly observed.
DOI: https://doi.org/10.7554/eLife.35224.008

in soluble protein depleted nucleoli could be found in their oil isolated counterparts. This observation indicates that soluble protein depletion does not result in irreversible aggregation as they still could serve as exchange donors. Next, we asked if soluble protein depleted nucleoli were still capable of receiving additional protein. To this end, we merged unlabeled soluble protein depleted nuclei with dually labeled oil isolated nuclei. Depleted nucleoli readily accumulated labeled proteins. This observation indicates that soluble protein depletion does not irreversibly alter the ability of nucleoli to accumulate additional proteins in their aggregates (*Figure 3B*).

In the first two rows of *Figure 3B* the fusion of soluble protein depleted nuclei with oil isolated nuclei tested transfer in the presence of approximately half the normal concentration of both soluble protein and ATP. The last row of *Figure 3B* tests whether fusion of two soluble protein depleted nuclei would behave the same way. We mixed labeled and unlabeled soluble protein depleted nuclei. Surprisingly, we could detect a faint exchange of GFP-NPM1 between soluble protein depleted nuclei. RFP-Fbl also exchanged between soluble protein depleted nucleoli, but transfer was even lower than that seen for GFP-Npm1 (bottom, *Figure 3B*). Thus, the large reduction of soluble nuclear proteins and small molecules, including ATP, compromised but did not eliminate protein exchange between nucleolar aggregates. The results show qualitatively that exchange is occurring. A quantitative approach would need to control or know the concentration of both the fluorescent reporter proteins and their endogenous pools, the distance from particle to particle(*Figure 3E*)and differences in the viscosity of the shared environment of the nuclei after fusion.

We noticed in the experiments described in *Figure 2* that if nucleoli were undergoing fusion at the time of aqueous isolation and soluble protein depletion that they fail to resolve into their normal spherical shape (*Figure 2C*). Ex vivo, others have reported a similar stiffened nucleolar phenotype upon ATP depletion (*Brangwynne et al., 2011*). Readdition of physiologic levels of ATP failed to rescue this phenotype (*Figure 2C*). However, fusion of a soluble protein depleted nuclei into an intact oil isolated nucleus rescued the ability of nucleoli to undergo fusion and return to a spherical shape (*Figure 3D*). This suggests that partial restoration of normal soluble nuclear factor(s) and ATP levels promotes a more normal rounded fusion product.

Mixing of soluble protein depleted and oil isolated nuclei revealed that soluble protein depletion does not irreversibly confer resistance to disaggregation, abolish the ability for resolution of stalled nucleolar fusion events, or prevent exchange between pre-existing aggregates.

## Nuclear protein aggregate disassembly is enhanced by ATP and a diffusible factor

The experiments above suggests that soluble nuclear proteins are needed for normal maintenance of nucleolar aggregates. Work by Wuhr et al. (*Wühr et al., 2015*) and our own data (*Figure 1*) demonstrate that *Xenopus* oocyte nuclei contain a full complement of proteins and other components immediately after isolation. If a diffusible factor is required for ATP dependent solubilization, protein aggregates might respond to ATP if nuclei are isolated and tested immediately.

We isolated nuclei from GFP-Npm1 and RFP-Fbl expressing stage V-VI oocytes and immediately transferred them to ISB supplemented with 8 mM ATP. This allowed us to assay aggregate solubilization by beginning the assay with at least a partial complement of nuclear components. During the assay soluble nuclear proteins, as well as Npm1 and Fbl that leave their coalesced state, are prone to dilution by equilibration with the surrounding buffer. Fluorescent microscopy of isolated nuclei 15 min after treatment revealed compartment specific changes in aggregated nucleolar proteins not observed in assays using soluble protein depleted nuclei. Specifically, aggregated GFP-Npm1 was absent or limited to a thin layer surrounding the visably intact RFP-Fbl containing dense fibrillar component (middle, *Figure 4A*). We also note that treatment with AMP-PNP alone has limited effect on either aggregated Npm1-GFP or Fbl-RFP (bottom, *Figure 4A*), indicating it cannot replace hydrolyzable ATP in this assay.

We next performed a dilution series to monitor ATP mediated aggregate solubilization (*Figure 4B*). Treatment with 8 mM ATP resulted in near complete granular component solubilization, indicated by the dispersion of GFP-Npm1 (*Figure 4B–C*). In contrast, RFP-Fbl positive particles from 2 to 6 mM ATP treated nuclei were enveloped with a thin layer of GFP-Npm1 (middle, *Figure 4A*), generally absent following 8 mM ATP treatment. Loss of RFP-Fbl was also observed following treatment with 2–8 mM ATP. However, this loss was not as complete as that observed with GFP-Npm1.

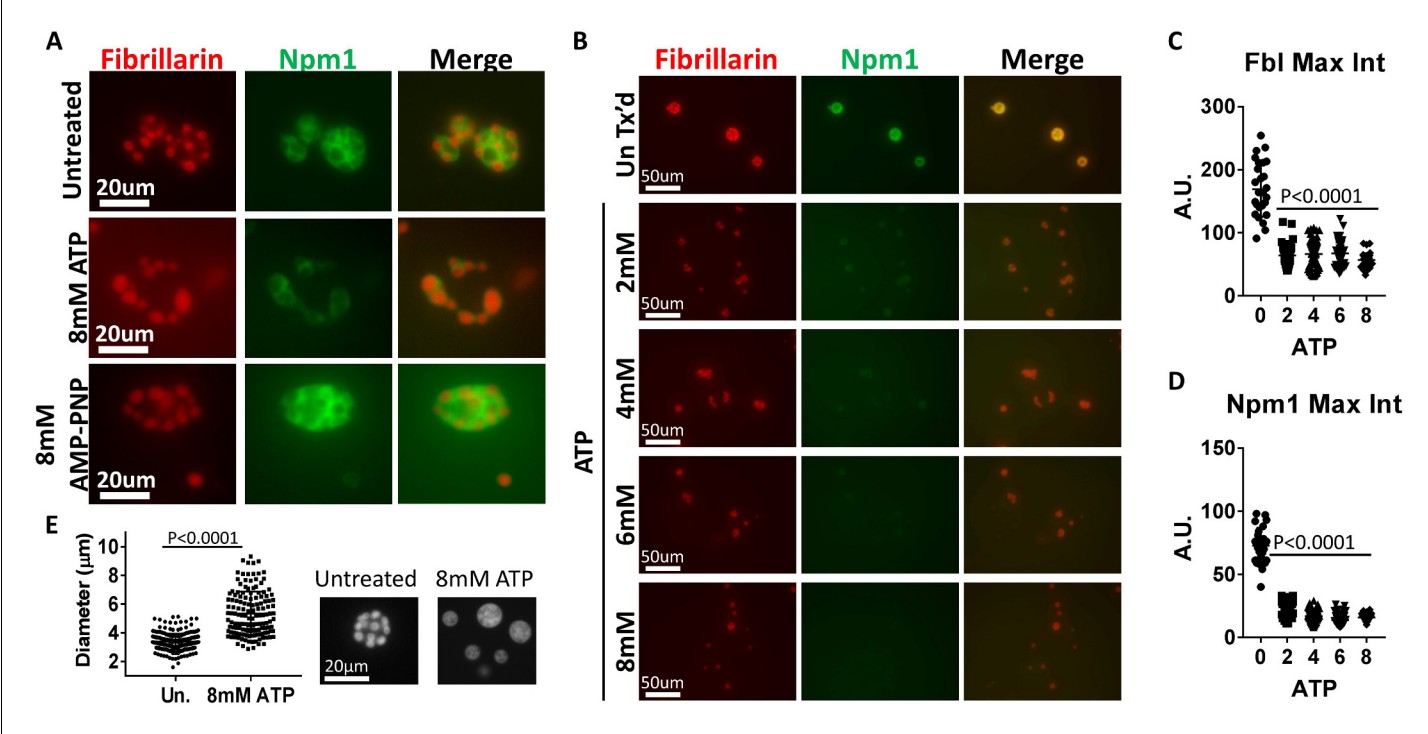

**Figure 4.** ATP hydrolysis and a soluble factor(s) are required for nucleolar remodeling. (**A**) Nuclei were isolated from RFP-Fbl and GFP-Npm1 expressing oocytes, immediately transferred into ISB alone (top), ISB supplemented with 8 mM ATP (middle), or ISB supplemented with 8 mM AMP-PNP (bottom), and imaged. (**B**) Freshly isolated nuclei were transferred into ISB supplemented with increasing amounts of ATP. (**C–D**) Relative maximum fluorescence intensity of GFP-NPM1 and RFP-FBL co-expressing nuclei incubated in ISB (untreated) or ISB supplemented with 0–8 mM ATP. (**E**) Diameter of RFP-Fbl positive Dense Fibrillar Component following incubation of freshly isolated nuclei in increasing concentrations of ATP. To test the effect of soluble nuclear proteins, nuclei in (**A–E**) were isolated and immediately placed into ISB with or without indicated supplementation. Images in (**A–D**) are representative of at least three independent experiments. Data in (**E**) represents 298 independent untreated and 144 independent 8 mM ATP treated RFP-Fbl foci. Data in (**C–D**) is from a single experiment with at least 25 nucleoli per data point.

DOI: https://doi.org/10.7554/eLife.35224.009

Additionally, aggregated RFP-Fbl foci from nuclei treated with higher levels of ATP appeared larger than those treated with lower levels (*Figure 4C*). The diameter of untreated and 8 mM ATP treated RFP-Fbl foci were measured; this analysis revealed an increased diameter of aggregated RFP-Fbl loci following 8 mM ATP treatment. This observation is consistent with gravity dependent spreading of a liquid droplet on top of a glass slide, demonstrating the droplet like nature of the Fbl aggregate is maintained following ATP treatment (*Figure 4C*).

In vitro trials using purified protein aggregates indicated that ATP, GTP and AMP-PNP were similarly effective as hydrotropes. (*Patel et al., 2017*) When soluble protein depleted nuclei (identified by phalloidin staining in *Figure 5*) and freshly isolated nuclei were added to the same reaction chamber containing ISB supplemented with 8 mM ATP or 8 mM GTP we examined whether nuclear aggregates were present by adding thioflavin T (an amyloid aggregate stain) to the chamber. Both nucleotides mediate the loss of thioflavin T positive aggregates in freshly isolated nuclei while neither changed thioflavin T aggregate staining in soluble protein depleted nuclei (*Figure 5A–C*).

These findings indicate that a diffusible factor or factors are critical for rapid solubilization of aggregated nucleolar components in combination with a hydrolysable ATP or GTP. Non-hydrolysable AMP-PNP was not sufficient in these assays. The response of the two protein reporters was not identical, with Npm1 easier to visualize as having left the coalesced state. At intermediate ATP concentration we observed retention of a thin layer of granular component surrounding an more ATP resistant dense fibrillar component core. This arrangement of the monitored LLPS protens is reminiscent of what Ferric and colleagues demonstrated highlighting the role of surface tension in driving the separation and arrangement of nucleolar compartments. (*Feric et al., 2016*) This observation is

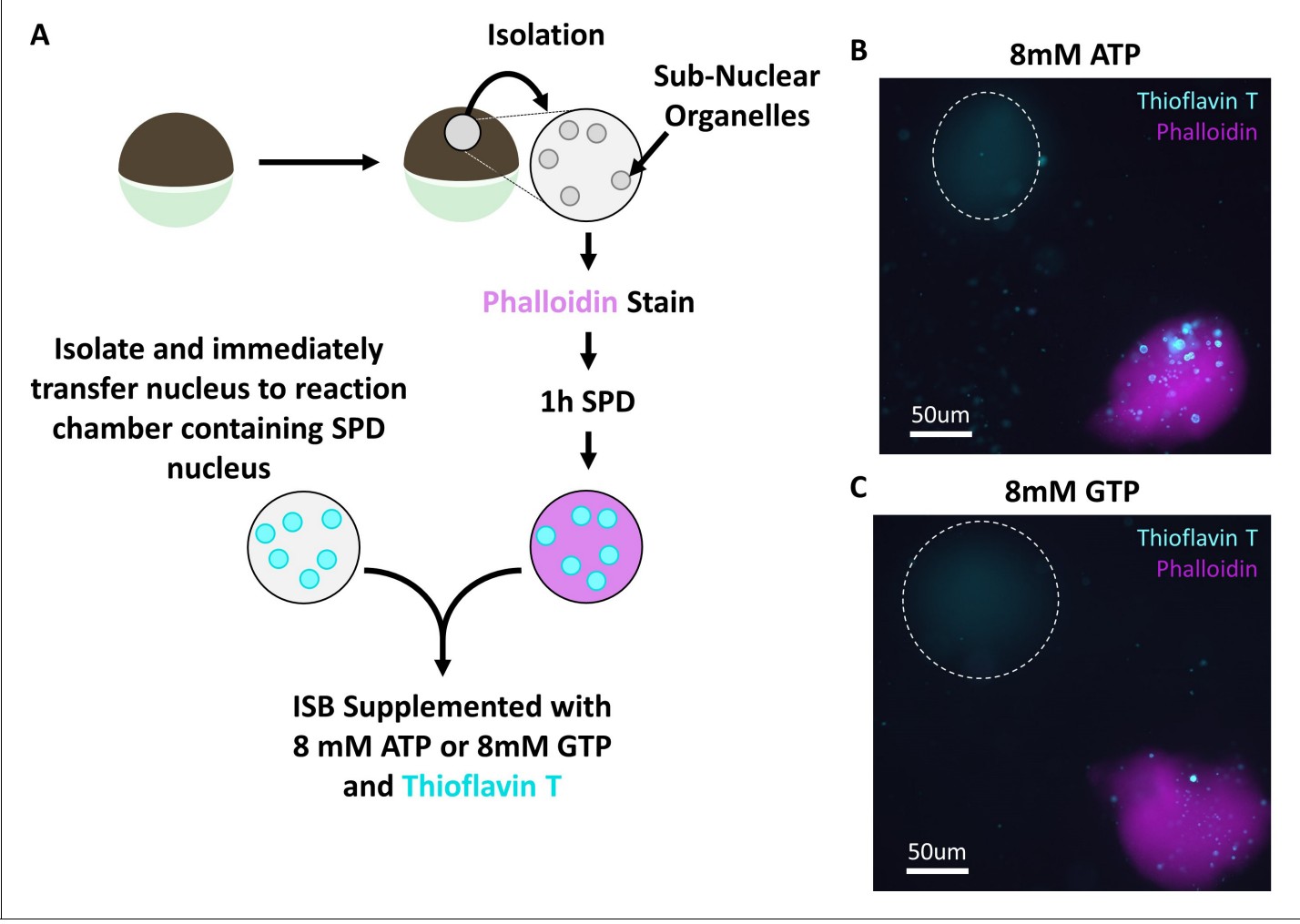

**Figure 5.** Treatment with 8 mM GTP replicates ATP results. (**A**) Nuclei were stained with stained with Alexa Fluor 568 conjugated phalloidin (Molecular Probes), depleted of soluble proteins for 1 hr and then placed into a reaction chamber with ISB supplemented with thioflavin T and 8 mM ATP or 8 mM GTP. A second nucleus was subsequently isolated and immediately added to the same reaction chamber. Isolation and immediate addition to ISB ensures transfer of soluble proteins. Images were acquired 15 min after addition of freshly isolated nucleus.
DOI: https://doi.org/10.7554/eLife.35224.010

consistent with there being heterogeneity within nucleolar compartments that differ in response to the presence of hydrolyzable ATP.

## In situ and ex vivo nucleolar aggregate responses to ATP, AMP-PNP

We tested if an artificially high level of ATP or AMP-PNP might reveal hydrotropic solubilization of nucleolar aggregates. Nuclei from GFP-Npm1and RFP-Fbl co-expressing oocytes were isolated under mineral oil to retain soluble protein and nuclear contents. Nuclei were then injected with 9.2 nL (an approximately equal volume) of 50 mM ATP or AMP-PNP, to yield a final nucleotide concentration of 25 mM. 25 mM is more than a 4-fold increase over the physiologic ATP concentration and nearly six times the minimum hydrotropic concentration of both ATP and AMP-PNP reported in vitro (*Patel et al., 2017*; *Miller and Horowitz, 1986*).

Nucleoli from AMP-PNP injected nuclei remained grossly unchanged (middle compared to top, *Figure 6B*). In contrast, ATP supplementation resulted in rapid loss of aggregated GFP-Npm1 (bottom, *Figure 6B* and *Figure 6C*). Nucleoli commonly contain multiple RFP-Fbl positive foci that are part of the dense fibrillar component (*Figure 2A*). Following in vivo ATP supplementation, this arrangement was rarely observed. Instead, RFP-Fbl positive foci dispersed into individual particles

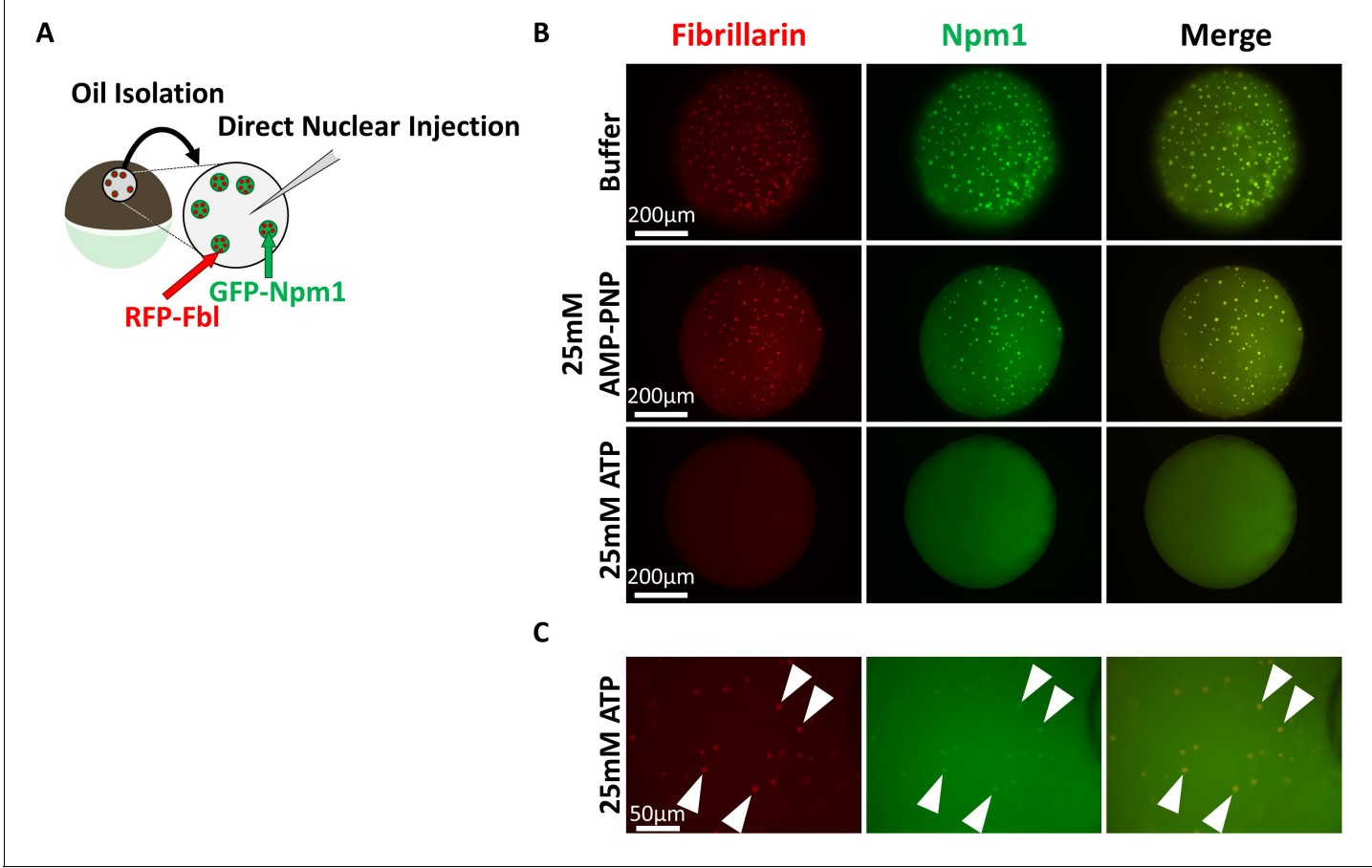

**Figure 6.** Super physiologic ATP promotes Nucleolar Disassembly Ex Vivo but AMPPNP does not. (**A**) Nuclei from GFP-Npm1 and RFP-Fbl expressing oocytes were isolated under oil to maintain nuclear contents and were injected with an ~equal vol (9.2 nL) of buffer (Top), 50 mM AMP-PNP (middle), or 50 mM ATP (Bottom). (**B**) High magnification image of 50 mM ATP injected nucleus. Arrowheads indicate resistant RFP-Fbl positive puncta. Images in (**B–C**) are representative of 9 (buffer), 12 (AMP-PNP), or 18 (ATP) biological replicates.

DOI: https://doi.org/10.7554/eLife.35224.011

(arrowheads, *Figure 6C*). Some of these particles were surrounded by a thin veil of GFP-Npm1, reprising the stable interaction observed ex vivo (middle, *Figure 4A*). In this assay, ATP can partici-pate as both an energy source and hydrotrope to promote solubilization of aggregated proteins. In contrast, AMP-PNP supplementation will likely have an inhibitory effect on energy dependent reac-tions that use ATP while maintaining the potential for hydrotropic solubilization. Even at high con-centrations AMP-PNP did not mimic ATP, suggesting hydrotropic properties of the molecule in vivo is not sufficient.

Even dilute concentrations of soluble GFP-Npm1 and RFP-Fbl are capable of being recruited into pre-existing aggregates (*Figure 3B*) under conditions of sub physiologic ATP levels. Others have documented this dynamic exchange in physiologic or near physiologic ATP concentration (*Brangwynne et al., 2011*; *Feric et al., 2016*) in nuclei isolated under oil to preserve a pro-aggrega-tion environment. However, supplementation with 25 mM ATP inhibits re-aggregation of GFP-NPM1. This is consistent with the need to overcome the hydrotropic action of ATP to establish the normal equilibrium status of protein recruitment into ribonucleoprotein aggregates such as nucleoli.

## AMP-PNP supplementation of a Sub-Minimum hydrotropic concentration of ATP promotes aggregate solubilization

Treatment of freshly isolated nuclei with 2 mM ATP replicated many of the changes seen with higher concentrations of ATP (*Figure 4B–D*). When we added only 1 mM ATP our results were similar to

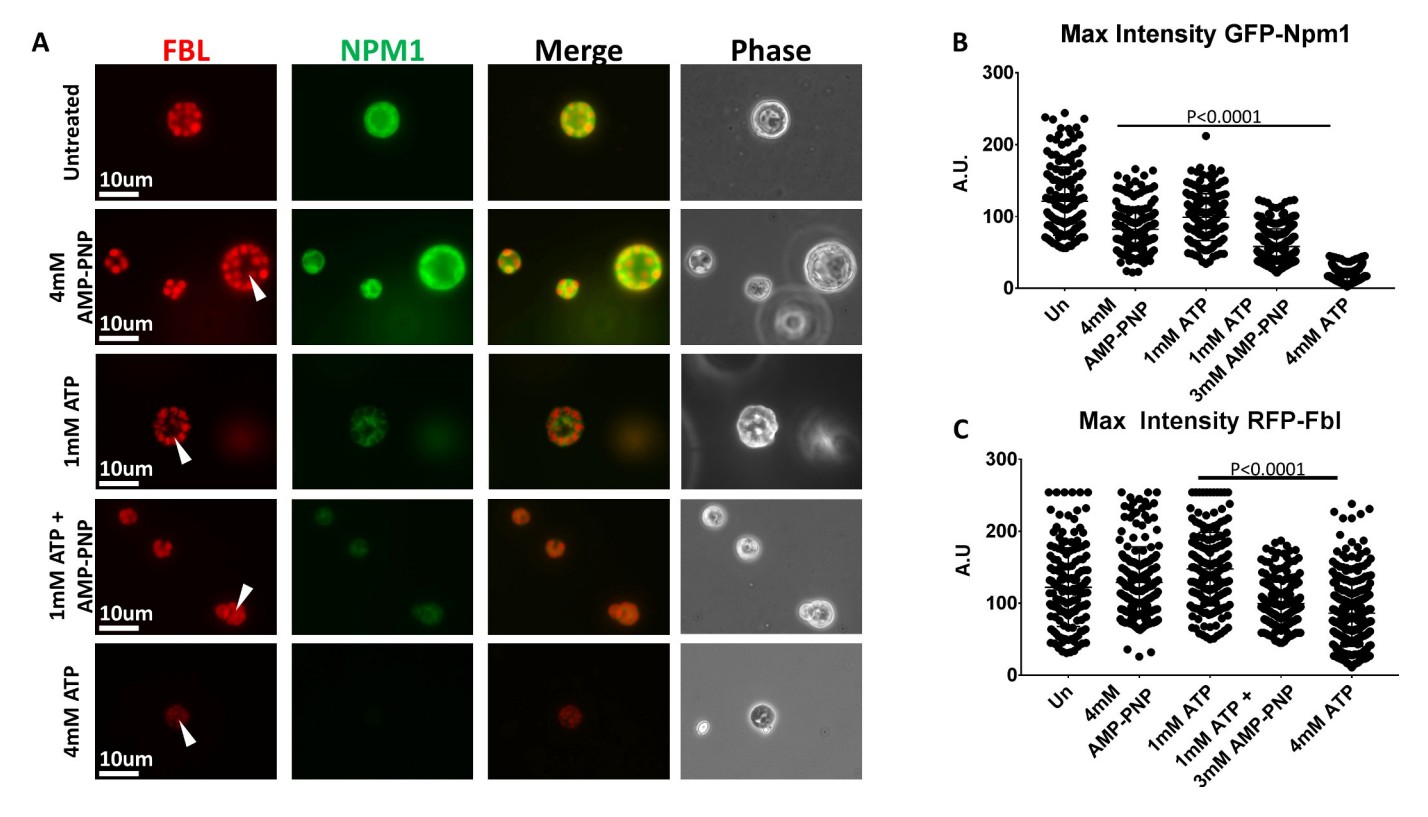

**Figure 7.** Hydrotropic Solubilization Contributes to Nucleolar Protein Dynamics. (**A**) Oocyte nuclei were isolated from RFP-Fbl and GFP-Npm1 expressing oocytes and immediately transferred into ISB alone (top) or ISB supplemented with ATP, AMP-PNP, or both nucleotides and imaged. (**B**) When compared to untreated controls, maximum fluorescence intensity measurements for GFP-Npm1 and C) RFP-Fbl reveals loss of signal intensity after 1 mM ATP + 3 mM AMP-PNP or 4 mM ATP supplementation. Images in (**A**) are representative of 3 independent experiments in which maximum fluorescence intensity of GFP-Npm1 (**B**) and RFP-Fbl (**C**) from 155 to 264 nucleoli was quantified, pooled, and analyzed.

DOI: https://doi.org/10.7554/eLife.35224.012

that we observed using 4 mM AMP-PNP(*Figure 7*). The reported $K_{M\ ATP}$ of protein and lipid kinases is rarely over 1 mM and the reported $K_{M\ ATP}$ of members of the yeast Hsp70 family of chaperones is less than 300 µM (*Knight and Shokat, 2005*; *Lopez-Buesa et al., 1998*), so we wondered whether at least a portion of the observed solubilization in earlier studies might be attributed to the hydrotropic action of ATP. We tested this premise by supplementation with another hydrotrope, non-hydrolysable AMP-PNP. We treated freshly isolated nuclei with 1 mM ATP, 1 mM ATP supplemented with 3 mM AMP-PNP, 4 mM ATP, and 4 mM AMP-PNP (*Figure 7A*).

In this assay treatment with 4 mM AMP-PNP or 1 mM ATP had a very modest effect on aggregated GFP-Npm1, and no decernible effect on aggregated RFP-Fbl. In contrast, and similar to our previous results (*Figure 4B–D*), incubation in 4 mM ATP resulted in solubilization of the granular component and partial solubilization of the dense fibrillar component. Supplementation of 1 mM ATP with 3 mM AMP-PNP demonstrated a synergistic effect (*Figure 7B–C*).

The ability of AMP-PNP to partially solubilize aggregated GFP-NPM1 and act synergistically with 1 mM ATP to partially solubilize aggregated RFP-Fbl demonstrates that it can act a hydrotrope in situ when the system is appropriately primed. More importantly, these observations indicate ATP may have more than one role in the regulation of endogenous protein aggregates. They support a model of nucleolar solubilization, wherein an energy dependent process may sensitize aggregated proteins to energy independent solubilization via a biological hydrotrope such as ATP.

# Sensitization of nucleolar compartments to the hydrotropic action of ATP using RNase A

RNA is an essential nucleolar component. It is required for LLPS of recombinant nucleolar proteins and for maintenance of nucleolar contents ex vivo (*Feric et al., 2016*; *Hayes and Weeks, 2016*). We have previously shown that maintenance of amyloid-like nucleolar organization and localization of proteins, like nucleolin, to nucleoli are sensitive to RNA depletion (*Hayes and Weeks, 2016*).

We tested the effect of RNA depletion on the sensitivity of Npm1 and Fbl to the hydrotropic action of ATP and AMP-PNP by pretreating with RNase A. Soluble protein depleted nuclei labeled with GFP-Npm1 and RFP-Fbl were prepared in buffer alone, or buffer supplemented with 1 mg/mL RNase A. These nuclei were subsequently transferred into ISB or ISB supplemented with 8 mM ATP or 8 mM AMP-PNP (*Figure 8A*).

Consistent with our previous results, RNase A treatment alone resulted in a decrease in the maximum fluorescent intensity of GFP-Npm1 (*Hayes and Weeks, 2016*). There was further loss of GFP-Npm1 when RNA depleted nuclei were treated with 8 mM ATP or AMP-PNP (*Figure 8C*). Treatment of RNA depleted nuclei with either compound induced mixing of previously phase separated nucleolar compartments (*Figure 8B*). This is consistent with the findings of Ferric et al. (*Feric et al., 2016*) demonstrating the RNA binding domains of each protein help to establish separate subcompartments in the nucleoli. After RNase A treatment either 8 mM ATP or AMP-PNP alone resulted in moderate loss of GFP-Npm1 (*Figure 8C*).

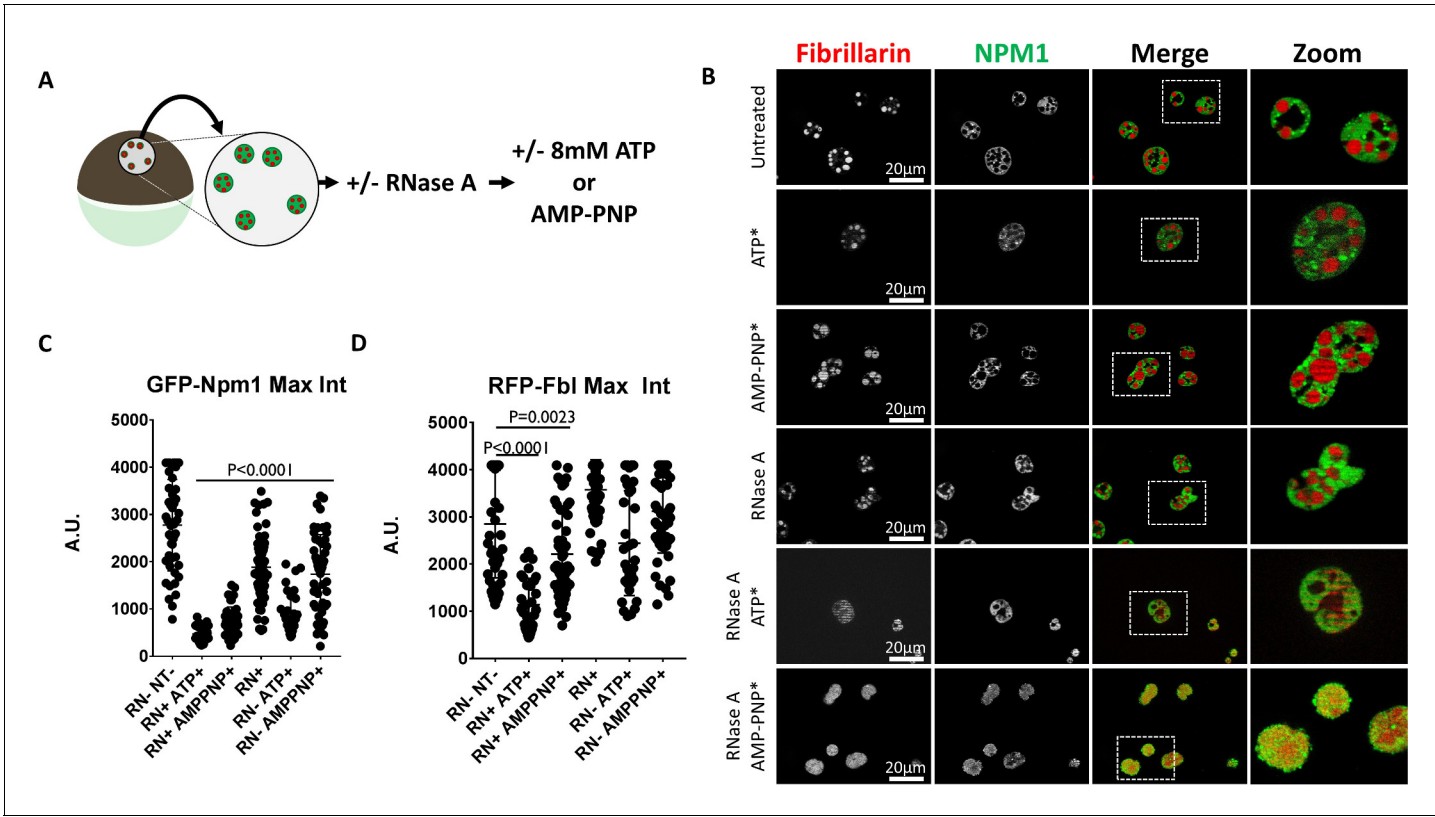

**Figure 8.** RNase Treatment Sensitizes Nucleoli to Hydrotropic Solubilization. (**A**) Nuclei from GFP-Npm1 and RFP-Fbl expressing oocytes were isolated, transferred into OR2 or OR2 supplemented with 1 mg/mL RNase A, and incubated for 1 hr prior to nucleotide treatment. (**B**) Optical sections through soluble protein depletion and RNase A treated soluble protein depleted nuclei left untreated (left), treated with 8 mM ATP (middle), or 8 mM AMP-PNP (right). (**C–D**) Intensity of GFP-Npm1 and RFP-Fbl. Data in (**B–D**) is representative of three independent experiments and three independent frogs. (**C–D**) represent a single experiment containing 3–6 biological replicates per treatment group and 129–205 individual nucleoli.
DOI: https://doi.org/10.7554/eLife.35224.013

In contrast, the stability of RFP-Fbl is more variable following the same treatments. RNA depletion followed by treatment with ATP or AMP-PNP were the only treatments which reliably resulted in loss of RFP-Fbl (*Figure 8D*).

The observation that ATP and AMP-PNP can solubilize aggregated nucleolar proteins, and induce mixing of distinct phase separated compartments, indicates that ATP can act as a hydrotrope to solubilize naturally aggregated proteins. Furthermore, RNA is also implicated as a mediator of nucleolar stabilization that confers resistance to ATP mediated hydrotropic solubilization.

## Discussion

Conditions that promote formation of protein aggregates have been the focus of innumerable studies, identifying contributions by amino acid sequence, protein concentration and modification and co-aggregates. Biological aggregates are in equilibrium with pools of non-aggregated protein, but are conditionally solubilized in response to environment, cell cycle, and other influences. Thus, the solubilization of protein aggregates is an essential process. Aggregate deconstruction has classically been attributed to energy dependent enzymatic reactions involving chaperones, such as Hsp70, and disaggregases like Hsp104. In in vitro studies using purified proteins, Patel and colleagues (*Patel et al., 2017*) reported that ATP can function as a hydrotrope to prevent or reverse protein aggregation. However, the ability of ATP to act as a hydrotrope in the context of endogenous protein aggregates was not investigated. Using isolated *Xenopus* oocyte nuclei, we dissected the ability of ATP to act as a hydrotrope to non-specifically solubilize endogenous protein aggregates or as an energy source to power enzymatic disaggregation.

The rapid loss of soluble proteins from nuclei isolated in aqueous buffer has been problematic for experiments requiring an intact nucleoplasm or full complement of nuclear factors (*Wühr et al., 2015*; *Carroll and Lehman, 1991*; *Gardner et al., 2012*). We took advantage of this phenomenon to identify proteins which are stably associated with endogenous nuclear aggregates. Mass spectrometric analysis of retained complexes revealed ~120 proteins, accounting for ~10% of the total nuclear protein mass. Within this selective proteome are proteins that are part of non-membrane bound nuclear structures, RNA binding proteins, and proteins that are capable of forming aggregates (*Kato et al., 2012*). While our study focuses on the dynamics of aggregated nucleolar proteins, the aggregate enriched nuclear proteome generated provides a starting point for an expanded analysis of other nuclear structures.

The nucleolus is a non-membrane bound sub-nuclear organelle with verified protein aggregates. The nucleolus has well characterized domains and signature proteins within each domain have been shown to form LLPS aggregates (*Feric et al., 2016*). Of these, the best characterized are Nucleophosmin (Npm1) and Fibrillarin (Fbl), both of which were identified within our proteomic analysis.

We used fluorescently labeled variants GFP-Npm1 and RFP-Fbl (*Feric et al., 2016*), to investigate the role ATP plays during endogenous aggregate solubilization. We took advantage of being able to isolate nuclei in three distinct states. (1) Isolation of nuclei in OR2 buffer followed by one hour incubation in OR2 creates nuclei that have lost 90% of total nuclear proteins but retain aggregated proteins. We refer to these nuclei as soluble protein depleted. These nuclei will rapidly equilibrate small molecules, like ATP, to match the small molecule concentration of the assay buffer. (2) Isolation in OR2, and immediate placement in ISB supplemented with the desired concentration of nucleotide and immediately assayed. These nuclei will have at least a partial complement of soluble nuclear proteins in addition to aggregated complexes as long as they are assayed quickly (typically within 15 min of isolation). The more rapid exchange of small molecules like ATP allow manipulation of their concentration by addition to ISB. We refer to these as in situ nuclei. (3) Nuclei can be isolated under oil. Oil isolated nuclei retain endogenous levels of soluble proteins, aggregated proteins and small molecules. We refer to these nuclei as oil isolated or ex vivo.

Npm1 and Fbl nucleolar aggregates assayed in nuclei that were depleted of soluble nuclear proteins remain stable when treated with 8 mM ATP or AMP-PNP. Although 8 mM is slightly higher than the reported in vivo concentration of frog oocyte ATP ($6.2 \pm 0.7$ mM) (*Miller and Horowitz, 1986*), 8 mM ATP, GTP or AMP-PNP was effective when used by Patel et al. (*Patel et al., 2017*) for hydrotropic solubilization of purified protein aggregates in vitro. The failure to respond to hydrotropic conditions lead to testing for conditions that would lead to changes in the Npm1 and Fbl coalesced in nucleoli.

The nucleolar Npm1 and Fbl had not been rendered permanently aggregated. Fusion of soluble protein depleted nuclei with oil isolated nuclei showed that nucleoli from soluble protein depleted nuclei can act as both donor and acceptor of aggregation prone proteins such as Npm1 and Fbl. The exchange of proteins between nuceloli has been noted by others and is a reminder that participation in an aggregate is not necessarily a static state (*Phair and Misteli, 2000*). The fusion of two soluble protein depleted nuclei had low but decernible exchange of Npm1 and Fbl. Exchange activity was enhanced in the presence of soluble nuclear proteins and approximately half normal nuclear levels of ATP provided when oil isolated nuclei were fused with soluble protein depleted nuclei. We note that in these assays Npm1 seemed to be more active in exchange than Fbl. This is consistent with previous work emphasizing that aggregated protein complexes may exhibit different properties (*Feric et al., 2016*; *Kato et al., 2012*; *Schwartz et al., 2013*).

Banini and colleagues (*Banani et al., 2016*), proposed a model where client proteins aggregate around a more stable scaffold. In this context, Npm1 likely falls into the client category. The relative resistance of RFP-Fbl to solubilization suggests that it may contribute to nucleolar structure as a scaffold. Fbl is an RNA binding protein that demonstrates viscoelastic properties following phase separation in vitro, indicating Fbl may assume LLPS droplet and polymer structure. (*Feric et al., 2016*) Consistent with our previous work, this observation suggests that at least a subset of phase-separated Fbl may exist within an amyloid fiber conformation that could contribute to maintenance of a nucleolar scaffold.

The nuclear fusion experiment indicated that even half the levels of soluble nuclear proteins and small molecules allowed enhanced ability of proteins to join and leave their aggregated state. Using in situ conditions, by immediately assaying after isolation, we were able to show that 8 mM ATP or GTP, but not AMP-PNP leads to disruption of aggregates. We note that although GTP works in this assay, the cellular concentration of GTP is estimated to be between one quarter to one tenth that of ATP. It is also important to note that many enzymes, for example Hsp70 (*Abravaya et al., 1992*), are able to use GTP as a substitute for ATP, although, based on cellular concentration, the normal substrate for these enzymes would be ATP. Immediately treating isolated nuclei with 8 mM ATP revealed distinct domain specific responses. Aggregated Npm1 is largely dispersed, however, a thin veil of ATP resistant Npm1 was often observed surrounding RFP-Fbl foci.This observation is consistent with a strong interaction of coalesced Npm1 at the granular component: dense fibrillar component interface. Feric et al. examined the physical properties of purified Fbl and Npm1 in an aqueous environment (*Feric et al., 2016*) and found that the interfacial tension between phases drives Npm1 to engulf Fbl. A phenomenon we also note here.

Although the fluorescence intensity from aggregated RFP-Fbl foci was decreased, we noted expansion of RFP-Fbl aggregates that settled to the bottom of the depression slide used for imaging. The decrease in fluorescent signal was consistent with RFP-Fbl leaving the aggregated state, while the spreading of the remaining signal was reminiscent of the liquid droplet properties described by others (*Brangwynne et al., 2011*; *Feric et al., 2016*).

When Npm1 and Fbl aggregates were assayed using in situ conditions 2–8 mM ATP supplementation effectively changes aggregate appearrance. A similar assay using 1 mM ATP showed little effect, however, a mix of 1 mM ATP with 3 mM AMP-PNP mimics the effect on Npm1 aggregates seen with 4 mM ATP. We interpret this as evidence that the process depends upon both a hydrolyzable form of ATP needed by soluble nuclear proteins and a hydrotropic effect that can be provided by the non-hydrolyzable analog AMP-PNP.

RNA has previously been shown to induce aggregate formation or contribute to aggregate maintenance (*Banani et al., 2016*; *Boke et al., 2016*; *Feric et al., 2016*; *Hayes and Weeks, 2016*; *Kato et al., 2012*; *Schwartz et al., 2013*). We investigated the contribution of RNA to maintenance of aggregate stability of Npm1 and Fbl in nucleoli by treating isolated nuclei with RNase A. We found that RNA depletion sensitized nucleoli to solubilization by ATP and the non-hydrolysable analog AMP-PNP thus removing the requirement for ATP hydrolysis. We suggest that RNase A treatment may mimic or substitute for the destabilization step provided by ATP hydrolysis. These findings implicate ATP dependent remodeling enzymes, such as chaperones, disaggregases, and RNA helicases, as possible mediators of aggregated protein turnover. Furthermore, our results suggest that differences between our observations and those made Patel et al. in vitro may be in part due to differences in the RNA content of the examined LLPS aggregates (*Patel et al., 2017*).

We propose a 2-step model for the in vivo solubilization of nucleolar aggregates that takes into account our experimental observations and suggest that this model may be relevant for many functional aggregates (*Figure 9*).

In the proposed model, an initial sensitization requires concentrations of ATP 1 mM or less and is mediated by a soluble factor (*Figures 2–6*) or detabilzation of a co-aggregate like RNA. We suggest a second step, aggregate solubilization, can be mediated by the energy independent hydrotropic activity of ATP (*Figures 7–8*), but requires a relatively high (2 mM or more) concentration. Assembly of nuclear aggregates could be a simple reversal of this model, where a stabilizing interaction, such as RNA binding or protein modification exceeds the capacity of ATP to maintain protein solubility in the absence of hydrolysis. Stabilization promotes the coalescence of proteins and other components inducing liquid-like phase separation. Once phase separated, particles can mature into a more stable form through limited formation of amyloid fibers, whose maintenance is dependent on continuous interaction with a stabilizing factor.

Invoking the ability of ATP to behave as a hydrotrope raises the interesting possibility that the equilibrium between the aggregated and free pool of aggregate forming proteins is affected by protein sequence, co-aggregates, or post-translational modification; but also by cellular compartment and local concentration of ATP. There are differences in published values, both for intracellular ATP levels and total ATP levels (*Patel et al., 2017*; *Miller and Horowitz, 1986*; *Traut, 1994*; *Harper et al., 1998*; *Imamura et al., 2009*; *Miyoshi et al., 2006*; *Wright et al., 2016*; *Ytting et al., 2012*). In the conditions tested here the concentration of ATP in the nucleus may favor solubilization unless a stabilizer is present or proteins are altered to promote aggregation. Alternatively, the 4 or 5-fold lower concentration reported in the cytosol may require proteins, once aggregated, to undergo active regulation to take them apart.

The ability to measure and observe temporospatial variability of ATP concentrations is becoming more feasible. New tools have been developed to document changes in ATP concentration in response to metabolic conditions within single cells, and regional changes to intracellular ATP concentrations in response to cellular morphology (*Imamura et al., 2009*; *Ytting et al., 2012*; *Yaginuma et al., 2014*; *Suzuki et al., 2015*). These changes ranged from a 10% decrease to an over three-fold increase in local concentration and can be transient or long lived. Another factor that may influence the local concentration of ATP is the aggregation of proteins themselves. Patel et al. (*Patel et al., 2017*) note that FUS aggregates formed in a solution containing fluorescently labeled ATP were enriched in ATP by fourfold over the solution concentration. Thus, local cellular environment, age, metabolic changes in ATP turnover, or free intracellular ATP likely play an underappreciated role in the biology of protein aggregation.

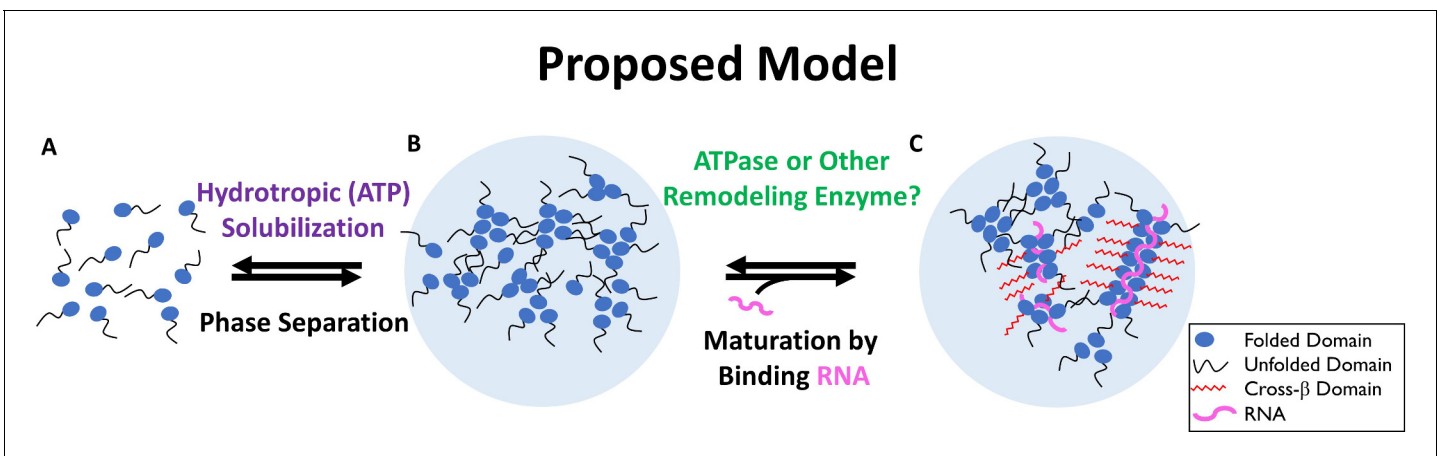

**Figure 9.** Two-Step Model of Nucleolar Aggregation and Disaggregation. Monomeric proteins self-associate driving phase separation. RNA binding stabilizes and matures aggregates, conferring resistance to hydrotropic solubilization. In contrast, ribonucleoprotein particles are destabilized by a diffusible ATPase or RNA depletion, sensitizing them to ATP mediated hydrotropic solubilization.
DOI: https://doi.org/10.7554/eLife.35224.014

Our observations add to the deliberation on the contribution ATP makes to the maintenance of protein solubility, functional protein aggregation and the regulation of non-membrane bound organelles. By showing that the hydrotropic properties of ATP can affect naturally occurring aggregates, we highlight the need to view endogenous levels of ATP within cellular compartments as more than a source of energy.

Our work also emphasizes the need to control for, or acknowledge, the influence of local concentrations when designing or interpreting experiments. This is readily apparent when looking at functional protein aggregation, where the average cellular contents or concentration of a molecule does not accurately reflect cellular conditions or the local protein environment. The local changes may also play an underappreciated role in the etiology of aggregate associated diseases, such as Alzheimer's and Parkinson's disease (*Harper et al., 1998*; *Miyoshi et al., 2006*).

## Materials and methods

### Oocyte isolation

J-strain *Xenopus laevis* frogs were obtained from the National Xenopus Resource at the Marine Biological Laboratory (RRID:SCR_013731) Woods Hole, MA). Ovarian tissue was aseptically removed 15 min following induction of anesthesia (ice bath supplemented with 0.1% Tricaine), placed into Oocyte Ringers ((OR2, 82.5 mM NaCl, 2.5 mM KCl, 1 mM $CaCl_2$, 1 mM $MgCl_2$, 1 mM $Na_2HPO_4$, 5 mM HEPES and NaOH to pH 7.8), and stored at 13°C. All protocols were approved by the University of Iowa Office of Animal Resources and Institutional Animal Care and Use Committee.

### In vitro transcription

Synthetic mRNAs were generated using a mMESSAGE mMACHINE kit (Ambion), following the manufacturer's instructions. Transcription templates were PCR amplified from RN3P (GFP-NLS) (*Zernicka-Goetz et al., 1996*); p3-dTomato (*Love et al., 2011*) (European *Xenopus* Resource Centre; Portsmouth, England); and pCPB055 (GFP:Npm1), pCPB057 (GFP-Fbl), and pCPB300 (RFP:Fbl) (*Knowles et al., 2014*; *Banani et al., 2016*). The primers used to generate templates for transcription reactions were: SP6 forward:

CATACATAGCATTTAGGTGACACTATAG;

M13 reverse:

CACACAGGAAACAGCTATGACC;

T3 Kozak.dTom: AAAAAAAATTAACCCTCAATAAAGGAGAGCCGCCACCATGGTGAGCAAGGCGAG; dTom.NLS; TTTTTTTTAGGGGTCTTCTACCTTTCTCTTCTTTTTTGGCTTGTACAGCTCGCCATGCC; RN3P.NLS TTTTTTTTTTTTTTAATCCAACTTCTTCTTCTTGGCGGCCGGTTTGTATAGTTCATCCATG; $NLSpA_{55}$;

TTTTTTTTTTTTTTTTTTTTTTTTTTTTTTTTTTTTTTTTTTTTTTTTTTTTTTTTTTTTTAATCC. Transcription templates for fluorescently labeled Npm1 an Fbl were kindly provided by Cliff Brangwynne (Princeton, NJ).

### Microinjection

Stage V-VI oocytes were manually defolliculated with watchmaker's forceps prior to injection of 1 ng of synthetic mRNAs in 10 nL using a glass needle paired with a Singer MK-1 (Somerset, England) micromanipulator and Inject+Matic injector (Geneva, Switzerland). Injection of 9.2 nL of solution into oil isolated nuclei ex vivo was performed using a MM33 micromanipulator mounted Nanoject II (Drummond). 10 nL of a 16 mM solution of Lucifer Yellow (Molecular Probes) was injected directly into oocyte nuclei for studies that examined movement of small molecules out of isolated nuclei.

### Endogenous aggregate enrichment and analysis

Nuclei from dTomato-NLS mRNA injected oocytes were harvested in OR2 and immediately transferred to a depression slide with 250 μL of OR2 supplemented with Thioflavin T (Sigma) to 50 μM. Lucifer Yellow experiments were performed in identical conditions without thioflavin T. For SDS-PAGE analysis, soluble proteins were depleted from isolated nuclei via incubation in ~5 mL of OR2 for the indicated times.

## Bulk nuclear isolation

Bulk nuclei were isolated using the method of Scalenghe et al., with slight modification. Briefly, ovarian tissue was enzymatically defolliculated (Collagenase A, Roche) and extensively washed in OR2. Stage V-VI oocytes were manually sorted and subsequently treated with 10 mg/mL Pronase (Sigma) for 30 min on ice with gentle agitation, then extensively washed in OR2 supplemented with 20 mg/mL BSA. Oocytes were lysed in a 60 mL syringe containing L-15 media supplemented with 0.2% NP-40 (three parts lysis medium to one part oocytes). With gentle agitation, intact nuclei float to the top of the syringe and are transferred into a 5 mL syringe using a three-way valve. Nuclei were purified form contaminating yolk and cytosolic content by sedimentation through L-15 containing 0.1M sucrose on ice. Nuclei were collected and transferred to 60 mM dishes containing 5 mL of OR2, where their envelopes were manually removed and soluble proteins were depleted. Samples were then processed for proteomic analysis.

## Proteomic analysis

Details on proteomic experiments are provided below and were based on protocols used in the Dovichi lab (*Peuchen et al., 2016*; *Wiśniewski et al., 2009*). Each sample of treated GV's was lysed and the proteome was extracted with an optimized buffer system of 6% SDS in Tris-HCL and protease inhibitor, followed by proteome cleanup and digestion with a filter aided sample preparation (FASP) method. The resulting digest was desalted and analyzed by nano-RPLC-ESI-MS/MS (90 min RPLC gradient, Q-Exactive mass spectrometer), followed by peptide and protein identification with MaxQuant software (RRID:SCR_014485) based on MS/MS spectra and label free quantification (LFQ) (*Cox et al., 2014*; *Cox and Mann, 2008*; *Cox et al., 2011*). All Proteomic data was deposited at MassIVE as noted below.

## SDS-PAGE, Western Blotting, and Antibodies

Protein samples were mixed with 2x Laemmli Sample Buffer (Bio Rad) supplemented with 2% β-mercaptoethanol (Amresco) and heated to 95°C for 10 min. Proteins were resolved on 10% Tris-Glycine gels (Bio Rad), and transferred to a nitrocellulose membrane (Bio Rad) where appropriate. Total protein was stained with Coomassie brilliant blue in gel, or Revert Total Protein stain (LI-cor) on membrane. Membranes were blocked in TBST (20 mM Tris, 137 mM NaCl, and 0.1% Tween 20) supplemented with 5% non-fat powdered milk, incubated with primary antibody (anti-GFP, 1:2000 dilution of ab6556, Abcam RRID:AB_305564) in blocking buffer overnight at 4°C with agitation, briefly washed with TBST, then transferred into blocking buffer containing HRP conjugated secondary antibody (ab97051, Abcam RRID:AB_10679369) for 1–2 hr at room temperature, and washed with TBST. Washed membranes were briefly incubated in Supersignal West Pico or Dura substrate (ThermoFisher) prior to imaging. Images were acquired with an Odyssey FC and quantified using Image Studio software (LI-Cor).

## Microscopy

Nuclei were mounted within a petrolatum (Vaseline) well containing the indicated buffer or a depression slide as noted previously. Images were acquired with an AxioPlan or ApoTome fluorescent microscope running Axio Vision software (RRID:SCR_002677)and an AxioCam MRm or AxioCam MrC5 camera (Zeiss). Images were pseudocolored and processed with FIJI (National Institutes of Health RRID:SCR_002285) or Photoshop (Adobe Systems Inc. RRID:SCR_014199)

## Nucleotides

ATP (Roche) was resuspended to a final concentration of 100 mM in 25 mM Tris and brought to pH seven with NaOH. AMP-PMP (Sigma) was diluted to 94 mM in OR2 and brought to a pH of 7. Both AMP-PNP and ATP were further diluted directly into in situ buffer (ISB) (20 mM HEPES, 100 mM KCl, 0.2 mM EDTA, 0.5 mM DTT, 20% Glycerol pH 7.4) or 25 mM Tris for ex vivo injection.

## Statistical analysis

Statistical analysis was performed with GraphPad Prism (GraphPad Software, USA RRID:SCR_002798). Where applicable, outlying data points were removed via ROUT analysis, and significance between treatment groups was determined by unpaired t-test.

## Proteome methods

### Reagents and materials

Bovine pancreas TPCK-treated trypsin, urea, ammonium bicarbonate ($NH_4HCO_3$), dithiothreitol (DTT), iodoacetamide (IAA), sodium dodecyl sulfated (SDS), and Tris buffer were purchased from Sigma−Aldrich (St. Louis, MO). Acetonitrile (ACN) and formic acid (FA) were purchased from Fisher Scientific (Pittsburgh, PA). Water was purchased from Honeywell Burdick and Jackson (Wicklow, Ireland). Complete, mini protease inhibitor cocktail (provided in EASYpacks) was purchased from Roche (Indianapolis, IN). The Microcon 30 kDA MWCO centrifugal filter units with regenerated cellulose membrane were purchased from Merck (Darmstadt, Germany).

### Proteome extraction and preparation

Each sample of nuclei was suspended in 750 μL of lysis buffer (6% SDS in Tris-HCl buffer) containing protease inhibitor, followed by cell lysis via sonication with a Branson Sonifier 250 (VWR Scientific, Batavia, IL) for 10 min on ice. Then, the lysate was centrifuged at 15,000 g for 10 min, and the supernatant was collected into a new Eppendorf tube. Next, 100 μg protein from each sample was denatured at 90°C for 30 min, reduced with DTT at 60°C for 1 hr, and then alkylated with IAA at room temperature for 30 min. A 30 kDa membrane was moistened using 100 μL of 100 mM $NH_4HCO_3$ and centrifuged for 15 min at 13,000 g. The alkylated sample was placed on the membrane and centrifuged for 40 min at 14,000 g. The membrane was washed three times with 200 μL of 8 M urea and centrifuged at 18,000 g for 25 min each time. A final wash with 100 μL of 20 mM $NH_4HCO_3$ was centrifuged for 10 min at 16,000 g. A tryptic digest of 1:25 trypsin:protein was completed by vortexing the samples for 10 min, followed by incubating at 37°C for 12 hr. FA was used to quench the digest.

After protein digestion, the peptides were collected via centrifuge. In order to improve the peptide recovery, we used 40 μL twice of 20 mM $NH_4HCO_3$ to wash the membrane again in order to collect the residual peptides on the membrane. Peptides from those two step collections were combined and lyophilized. After that, the peptide sample was redissolved to 1 μg/μL of 1% (v/v) FA, lyophilized per manufacturer instructions using SPE columns. The desalted same was again reconstituted to 1 μg/μL of 1% (v/v) FA for reversed-phase liquid chromatography (RPLC)-electrospray ionization (ESI)-tandem mass spectrometry (MS/MS) analysis.

## RPLC-ESI-MS/MS

A nanoACQUITY UltraPerformance LC (UPLC) system (Waters, Milford, MA, USA) was used for peptide separation. Buffer A (0.1% FA in water) and buffer B (0.1% FA in ACN) were used as mobile phases for gradient separation. The flow rate was 1 μL/min. Peptides were automatically loaded onto a commercial C18 reversed phase column (Waters, 100 μm × 100 mm, 1.7 μm particle, BEH130C18, column temperature 40°C) with 2% buffer B for 10 min, followed by 3-step gradient separation, 1 min from 2% to 8% B, 60 min to 30% B, 1 min to 80% B, and maintained at 80% B for 6 min. The column was equilibrated for 11 min with 2% B before analysis of the next sample. The eluted peptides from the C18 column were pumped through a capillary tip for electrospray, and analyzed by a Q-Exactive mass spectrometer (Thermo Fisher Scientific). The electrospray voltage was set at 1.8 kV. The ion transfer tube temperature was 300°C. Full MS scans were acquired from 350 to 1800 m/z in the Orbitrap mass analyzer at a resolution of 70,000 (at 200 m/z). The AGC target value was 3.00E + 06. The maximum injection time was 60 ms. Two microscan was used. The twelve most intense peaks with charge state ≥2 were isolated in the quadrupole with isolation window of 2.0 m/z units, NCE was set at 28%, and further analyzed in the Orbitrap mass analyzer with 35,000 resolution (at 200 m/z), 1.0E + 06 AGC target value, and 120 ms maximum injection time. The intensity threshold was 1.0E + 05 for triggering the peptide fragmentation. Dynamic exclusion was 40 s. Isotopes were excluded. For each RPLC-MS/MS run, 2 μL of the peptide sample from each blastomere was loaded onto the RPLC column for analysis.

## Data analysis

All the raw MS files were analyzed by MaxQuant version 1.5.3.30. MS/MS spectra were searched against the protein reference database from *Xenopus laevis* genome 9.1 (*Cox et al., 2014*, *2011*). The analysis included an initial search with a precursor mass tolerance of 20 ppm, main search precursor mass tolerance of 4.5 ppm and fragment mass tolerance of 20 ppm. The search included

enzyme was trypsin, variable modifications of oxidation (M), acetylation (protein N-terminal) and deamidation (NQ), and fixed modification of carbamidomethyl cysteine. Minimal peptide length was set to six amino acids and the maximum number of missed cleavages allowed was two. The false discovery rate (FDR) was set to 0.01 for both peptide and protein identifications. All proteins identified by the same sets of peptides were grouped, and reported as one protein group. Protein groups identified with at least one unique peptide were considered as positive hits. The protein group table was filtered to remove the identifications from the reverse database and common contaminants. 'Second peptides' and 'Match between runs' functions with default settings were employed for database searching. 'Label free quantification' function in the MaxQuant software was used with default settings.

## Data sets deposition

Proteomic data sets generated for this study are deposited at MassIVE data ftp://massive.ucsd.edu/MSV000081690/.

It is important to note that data sets for both effective and ineffective protocols for identification of aggregate and aggregate associated proteins are deposited. The data set obtained using the protocol indicated and designated 'Purple' included proteins we verified as retained and aggregated using both microscopy and western blot analysis. Other preparations identified fewer proteins and failed to identify known aggregates.

Rapid isolation and collection of manually dissected Xenopus oocyte nuclei has been used to describe the nuclear proteome of a vertebrate. Like those studies, we found that soluble proteins are easily depleted from manually isolated nuclei by simple extending the time between dissection and collection (*Figure 1*). Our initial attempts to characterize the aggregated nuclear proteome used the limited amount of material we were able to manually harvest in a reasonable amount of time. We next investigated the use of two different bulk nuclear isolation protocols. Date sets from these trials and the dataset used for the analysis in this manuscript are all deposited at MassIVE.

Our initial attempts at bulk nuclear isolation were modeled off the protocol described by *Lehman and Carroll (1993)*. Retained proteins from nuclei isolated in this manner did not replicate the pattern seen following soluble protein depletion of manually isolated nuclei. Therefore, we attempted further enrich for aggregate associated proteins by washing 'Carroll' isolated nuclei with OR2. In parallel experiments, we wanted to identify RNA dependent aggregate associated proteins. To do so, we washed bulk isolated nuclei with 10 mg/mL RNase A supplemented OR2 for 1 hr. These additional manipulations also failed to replicate the results of manually isolated and soluble protein depleted nuclei.

We next performed bulk nuclear isolation based on the protocol described by Scalenghe et al. (*Scalenghe et al., 1978*) SDS PAGE analysis of these samples (designated 'purple' in the deposited) more closely resembled the results of our manually isolated and soluble protein depleted nuclei. Therefore, we proceeded with this protocol, with two modifications. Following isolation, we incubated nuclei in OR2 for 1 hr, to more closely replicate our manual isolation protocol, and we manually removed nuclear envelopes to limit our analysis to intra nuclear aggregates. Nuclear envelopes were collected and included as a separate sample in our proteomic analysis.

In the deposited data sets samples designated IGS represent proteins recovered from fresh nuclei, nuclei depleted of soluble proteins are in data sets: Wash N3-10-1,Wash N3-10-2,Wash N63-10,OR2 and Purple Wash. Background controls are found in datasets Unknown Blank, SN1, SN6 and SN3.

We have included data sets from initial trials with four biological replicates examined either in duplicate or triplicate. Samples designations were 'Fresh' for whole nuclei, Pellet for retained proteins (including envelope proteins) and SN for proteins that left the nucleus during equilibration in OR2. In addition, a single biological replicate, analyzed in triplicate, of nuclei treated with RNaseA and of isolated nuclear envelope protein is included as part of the sample coded 'Purple'. They are deposited as part of the full analysis made on this biological replicate but await biological replicates prior to full analysis. A key with brief descriptions of the data sets can be found in *Table 1—source data 2*.

## Acknowledgements

The authors acknowledge the generosity of Dr. C Brangwynne (Princeton University) for plasmids encoding fluorescent proteins used in this study. Drs. J Fassler and B Phillips (University of Iowa) provided insight during the completion of this work. MH is a trainee in the Medical Scientist Training Program at the University of Iowa and the work is partial fulfillment of the requirements for his doctoral degree. Partial funding for these studies was from the Internal Research Funding Initiative at the University of Iowa to DW. NJD acknowledges support from the National Institute of Health (R01GM096767 and R01HD084399). EHP is funded by the National Science Foundation Graduate Research Fellowship (2015–2018).

## Additional information

### Funding

| Funder | Grant reference number | Author |
| --- | --- | --- |
| National Institute of General Medical Sciences | Medical Scientist Training Program | Michael H Hayes |
| National Science Foundation | Pre-doctoral Fellowship 2015-2018 | Elizabeth H Peuchen |
| National Institute of General Medical Sciences | R01GM096767 | Norman J Dovichi |
| National Institute of General Medical Sciences | R01HD084399 | Norman J Dovichi |
| University of Iowa | Internal Funding Initiative | Daniel L Weeks |

The funders had no role in study design, data collection and interpretation, or the decision to submit the work for publication.

### Author contributions

Michael H Hayes, Conceptualization, Data curation, Formal analysis, Validation, Investigation, Visualization, Methodology, Writing—original draft, Writing—review and editing; Elizabeth H Peuchen, Data curation, Formal analysis, Validation, Investigation, Methodology, Writing—review and editing; Norman J Dovichi, Resources, Data curation, Supervision, Funding acquisition, Writing—review and editing; Daniel L Weeks, Conceptualization, Resources, Supervision, Funding acquisition, Validation, Investigation, Visualization, Methodology, Writing—original draft, Project administration, Writing—review and editing

### Author ORCIDs

Daniel L Weeks http://orcid.org/0000-0002-4977-2410

### Ethics

Animal experimentation: This study was performed in accordance with the the Guide for the Care and Use of Laboratory Animals of the National Institutes of Health. All protocols were approved by the University of Iowa Office of Animal Resources and Institutional Animal Care and Use Committee (Animal Protocol #5041363-003 ) to minimize any animal suffering.

### Decision letter and Author response

Decision letter https://doi.org/10.7554/eLife.35224.019
Author response https://doi.org/10.7554/eLife.35224.020

## Additional files

### Supplementary files

• Transparent reporting form

DOI: https://doi.org/10.7554/eLife.35224.015

#### Data availability

Proteomic Data set access directions are in the Supplemental text: "Proteomic data sets generated for this study are deposited at MassIVE data ftp://massive.ucsd.edu/MSV000081690/."

The following dataset was generated:

| Author(s) | Year | Dataset title | Dataset URL | Database, license, and accessibility information |
|---|---|---|---|---|
| N Dovichei, E Peuchen | 2017 | Collaboration between Iowa and Notre Dame. Proteomic data run at Notre Dame | ftp://massive.ucsd.edu/MSV000081690/ | Publicly available at MassIVE data. |

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
