## [Decision Letter]

Thank you for submitting your article "Dual roles for ATP in the regulation of phase separated protein aggregates in *Xenopus* oocyte nucleoli" for consideration by *eLife*. Your article has been reviewed by three peer reviewers, one of whom is a member of our Board of Reviewing Editors, and the evaluation has been overseen by Didier Stainier as the Senior Editor. The reviewers have opted to remain anonymous.

The reviewers have discussed the reviews with one another and the Reviewing Editor has drafted this decision to help you prepare a revised submission.

Summary:

In this manuscript, Wells et al. build upon a recent finding from that ATP can function as a hydrotrope to increase protein solubility and investigate this function of ATP in nuclei isolated from *Xenopus* oocytes. The authors identified that nucleolar proteins selectively aggregate in nuclei extracted in aqueous buffers. The authors employ fluorescently tagged nucleolar proteins Npm1 (as a marker for granular component) and Fbl (as a marker for the fibrillar component of the nucleolus) and use fluorescence microscopy to measure the extent of protein aggregation in various extraction conditions. Notably, the authors find that ATP and a non-hydrolysable ATP analog, ANP-PNP have differential activities in solubilizing the nucleolar components.

The key claim in the paper is that the increased solubility of the nuclear aggregates in the presence of ATP arises from a combination of ATP acting as an energy source and as a hydrotrope. ATP consuming chaperones such as Hsp70 have been known to increase protein solubility both in vitro and in cells. This is interesting, as it suggests how the dual roles of ATP as a hydrotrope and as an energy source might function to keep proteins soluble.

While we are potentially interested in publishing the manuscript, we would like you to address the following controls and clarifications. In particular the paper would benefit from more focus on the problem to be solved and how it is being addressed. This paper makes an important point about the dual roles of ATP. As currently written, it contains a number of statements that are distracting from the message.

Essential revisions including issues that can be addressed by modifying the text:

1) The authors should investigate whether the observed effects arise due to salts that accompany ATP/ATP analogs by preparing solutions at identical salt compositions. For hydrotrope effect, the authors may use a mixture of adenosine base and phosphate salts as additional controls. Similarly, authors should directly show that ATP hydrolysis is essential for the proposed 'sensitization'.

2) The rationale for choosing different buffer conditions should be better articulated in the manuscript. The authors should show that the ISB/TB buffer does not lead to protein depletion.

3) The work would also benefit from careful quantitative analysis of microscopy data. The extent of exchange in Figure 3 or relocalization of proteins in Figures 4-7 should be quantified. Similarly, the time-scales of experiments with 'rapid isolation' could be better defined.

In particular:

a) please quantify Figure 4 as a dose response curve of Fibrillarin and Nucleoplasmin intensities in response to ATP and AMP-PNP concentrations.

b) Have a figure and dose response curve comparing the Fibrillarin and Nucleoplasmin intensities in response to ATP in the presence and absence of the diffusible factor.

4) The authors propose that there might be two species of NPM1 in the GC. However, this proposal does not come with any evidence. Pending any investigations of the conformational states of NPM1 to demonstrate that there indeed are two distinct species of NPM1, there might actually be a simpler explanation that derives from the work of Feric et al. They showed that NPM1 rounds up on hydrophobic surfaces and spreads on hydrophilic surfaces (see Brangwynne, Mitchison and Hyman, 2011). Therefore, a simpler way to think of the problem is in terms of interfacial tensions. If the interfacial tension between the GC and the solvent is lower than the interfacial tension between the GC and the DFC, both of which involve NPM1, then the thinning that is observed in the presence of non-hydrolyzable ATP would make sense. The observations would be entirely consistent with a capillary phenomenon rather than a two-species model. The authors would do well to include this as another plausible explanation for their observations regarding the impact of non-hydrolyzable ATP on the GC.

5) The discussion of LLPS in paper is misleading and somewhat unwarranted. The results that the authors describe do not necessarily pertain to liquid-liquid demixing. The authors should consider rewording the Introduction and Discussion sections.

6) The Introduction could use work. There are quite a few confusing statements, some which might end up being distracting and/or misleading. Considerable space is devoted to the topic of hydrotropes forming micelle-like structures. This may well be true for ATP and smaller hydrotropes, but Eastoe, Hatzopoulos, and Dowding (2011) is not the one to cite as it is focused on reviewing data for a very different class of hydrotropes. If the issue of ATP micellization needs to be raised – and it is not clear why this matters here – then this should come up in the Discussion, perhaps in a coherent discussion of linkage phenomena. And in doing so, please explain how/why micelle formation will lead to solubilization. Also, in general, the Introduction needs paring down and it should focus on describing the problem being solved (what), why it is important, and a short summary of main findings. As written, there are far too many distracting and confounding pronouncements.

7) The authors raise an interesting point in the Discussion and this pertains to the maintenance of concentration gradients and/or depots of ATP. The active liquid question and the existence of fields or loci of energy availability and its implications for organization of matter is of fundamental importance. Indeed, there have been elegant calcium/ATP labeling experiments in cells and these would be worth citing in the Discussion. Please see: http://www.jbc.org/content/274/19/13281.abstractand http://www.nature.com/srep/2014/141006/srep06522/full/srep06522.html. These papers and associated issues would be worth discussing in some detail.

---

## [Author Response]

Essential revisions including issues that can be addressed by modifying the text:1) The authors should investigate whether the observed effects arise due to salts that accompany ATP/ATP analogs by preparing solutions at identical salt compositions. For hydrotrope effect, the authors may use a mixture of adenosine base and phosphate salts as additional controls.

The reviewers make a good point, as salt composition and concentrations can certainly change protein interactions. In the comparison of ATP and AMP-PNP we are using the same salt conditions during direct comparison, and we have endeavored to make that clearer. Although the test with adenosine and phosphate salt could be done, we note that, AMP and ADP were directly tested by Patel et al., who concluded that although these compounds could mediate the hydrotropic effects of ATP, they require higher concentrations. Coupled with the disparity between normal cellular concentrations of AMP, ADP (or any other nucleotide triphosphate like GTP) and that of ATP we would suggest that control is no more meaningful that the comparison of ATP and AMP-PNP.

We include a direct look at GTP compared to ATP in an assay for retentions or loss of nuclear aggregates under the two conditions used for isolation of nuclei in aqueous buffer in the new Figure 5. We show that under conditions that retain soluble nuclear proteins that the thioflavin T detection of nucleolar aggregates is reduced. We discuss what that means with respect to potential action as an energy source and as a hydrotrope.

Similarly, authors should directly show that ATP hydrolysis is essential for the proposed 'sensitization'.

We understand the concern raised that we have not directly shown ATP hydrolysis but, the assays we employ would not be able to distinguish whether the hydrolysis is a direct, indirect or coincidental to the results. We can (and now do) replace the assertion that ATP is being hydrolysed with the more cautious indicator that AMP-PMP does not substitute for hydrolysable ATP.

2) The rationale for choosing different buffer conditions should be better articulated in the manuscript.

We have introduced the rationale in the Introduction.

The authors should show that the ISB/TB buffer does not lead to protein depletion.

First of all, we apologize for our error in not being consistent with our nomenclature. TB and ISB are the same buffer. We now uniformly use ISB. ISB does not result in the retention of nuclear proteins. We have altered the text to make this point clearer. The timing of treatment, immediate vs. 1h post isolation, that determines the protein/small molecule content of the sample. ISB buffer was used to more closely resemble the nuclear environment (e.g. KCl instead of NaCl) and facilitate enzymatic reactions. We note that the observations made in Figures 2C and 4A have made assaying in both OR2 and ISB, but for consistency in this work we present images captured from nuclei in ISB.

3) The work would also benefit from careful quantitative analysis of microscopy data. The extent of exchange in Figure 3 or relocalization of proteins in Figures 4-7 should be quantified. Similarly, the time-scales of experiments with 'rapid isolation' could be better defined.In particular:a) please quantify Figure 4 as a dose response curve of Fibrillarin and Nucleoplasmin intensities in response to ATP and AMP-PNP concentrations.

This quantification has been incorporated in graphs included in Figure 4 and is discussed.

b) Have a figure and dose response curve comparing the Fibrillarin and Nucleoplasmin intensities in response to ATP in the presence and absence of the diffusible factor.

A dose response curve has been included in Figure 4.

With respect to the quantification of protein relocalization: Our data clearly shows the transfer of protein from one nucleus to another, demonstrating our ability *to qualitatively assay* the ability of proteins to move between aggregated and soluble forms. Quantification of the kinetics of exchange, especially in comparison to previously data, would prove interesting. Trans-nuclear protein exchange is a diffusion limited phenomenon, which prevents us from making quantitative measurements in our system. For quantitative measurements to be possible, at least two criteria would need to be met: 1) Homogenous mixing of trans nuclei to control for the distance of diffusion, something we were unable to accomplish. 2) A constant concentration of labeled and unlabeled protein across mixing reactions. The amount of labeled protein within nuclei is dependent upon multiple factors. Some, such as mass of injected synthetic mRNA, we are able to control. Others, such as the volume of isolated nuclei, we are unable to control. We regret that these variables prevent us from accurately measuring protein exchange but feel that the qualitative observation can be interpreted and has value.

4) The authors propose that there might be two species of NPM1 in the GC. However, this proposal does not come with any evidence. Pending any investigations of the conformational states of NPM1 to demonstrate that there indeed are two distinct species of NPM1, there might actually be a simpler explanation that derives from the work of Feric et al. They showed that NPM1 rounds up on hydrophobic surfaces and spreads on hydrophilic surfaces (see Brangwynne, Mitchison and Hyman, 2011). Therefore, a simpler way to think of the problem is in terms of interfacial tensions. If the interfacial tension between the GC and the solvent is lower than the interfacial tension between the GC and the DFC, both of which involve NPM1, then the thinning that is observed in the presence of non-hydrolyzable ATP would make sense. The observations would be entirely consistent with a capillary phenomenon rather than a two-species model. The authors would do well to include this as another plausible explanation for their observations regarding the impact of non-hydrolyzable ATP on the GC.

The reviewers offer an alternative explanation of our observation that there may be a previously unappreciated conformation/species of NPM1. They pointed toward work carried out by Ferric et al. that made several observations on the interaction of NPM1 with hydrophobic and hydrophilic surfaces. Their point is valid and we have included the alternative explanation.

5) The discussion of LLPS in paper is misleading and somewhat unwarranted. The results that the authors describe do not necessarily pertain to liquid-liquid demixing. The authors should consider rewording the Introduction and Discussion sections.

We are sensitive to providing clarity with regard to what we are testing. We view LLPS as one form among many of protein aggregation. Both fibrillarin and nucleoplasmin (FBL and Npm1) have been featured in studies on LLPS so we feel it is necessary to point that out. We have added a reference (Berry et al.. 2018) that should be useful for readers interested in LLPS, including mixing and demixing. The data we present in the original Figure 7 (now Figure 8), as noted by the reviewer are not a direct test of mixing-demixing, but do suggest that a co-aggregate, in this case RNA, plays a role along with the proteins themselves and ATP in providing a suitable environment to maintain phase borders. As noted below, we have reworded both the Introduction and the Discussion.

6) The Introduction could use work. There are quite a few confusing statements, some which might end up being distracting and/or misleading. Considerable space is devoted to the topic of hydrotropes forming micelle-like structures. This may well be true for ATP and smaller hydrotropes, but Eastoe, Hatzopoulos, and Dowding (2011) is not the one to cite as it is focused on reviewing data for a very different class of hydrotropes. If the issue of ATP micellization needs to be raised – and it is not clear why this matters here – then this should come up in the Discussion, perhaps in a coherent discussion of linkage phenomena. And in doing so, please explain how/why micelle formation will lead to solubilization. Also, in general, the Introduction needs paring down and it should focus on describing the problem being solved (what), why it is important, and a short summary of main findings. As written, there are far too many distracting and confounding pronouncements.

The reviewer alerted us to the need to examine what is actually known about the structures that may be formed by hydrotropes. Although it is clear some hydrotropes do form micellular structure, it is not a universal feature. We have trimmed that section down to point readers to two other references that point out what self-associating structures have been reported for hydrotropes, including one recent commentary by Meyer and Voigt (2017) that was stimulated by the work of Patel et al. We appreciate the frank pronouncement regarding distracting and confound pronouncements and have made substantive changes to remedy the problem.

7) The authors raise an interesting point in the Discussion and this pertains to the maintenance of concentration gradients and/or depots of ATP. The active liquid question and the existence of fields or loci of energy availability and its implications for organization of matter is of fundamental importance. Indeed, there have been elegant calcium/ATP labeling experiments in cells and these would be worth citing in the Discussion. Please see: http://www.jbc.org/content/274/19/13281.abstractand http://www.nature.com/srep/2014/141006/srep06522/full/srep06522.html. These papers and associated issues would be worth discussing in some detail.

We agree and now point readers to these and a number of studies that may be relevant to both energy and hydrotropic roles for ATP (see Discussion, fifteenth paragraph).